# TOOLWEAVER: WEAVING COLLABORATIVE SEMANTICS FOR SCALABLE TOOL USE IN LARGE LANGUAGE MODELS

**Bowen Fang[1,2,3], Wen Ye[1,2], Yunyue Su[1], Jinghao Zhang[1], Qiang Liu[1], Yesheng Liu[2]**
**Jiabing Yang[1,2], Xin Sun[1] , Shu Wu[1][\*], Baole Wei[3,4][\*], Liang Wang[1]**
[1]New Laboratory of Pattern Recognition (NLPR),
Institute of Automation, Chinese Academy of Sciences (CASIA)
[2]School of Artificial Intelligence, University of Chinese Academy of Sciences
[3]Zhongguancun Academy
[4]Zhongguancun Institute of Artificial Intelligence
bwn.fang@gmail.com, shu.wu@nlpr.ia.ac.cn, weibaole@zgci.ac.cn

## ABSTRACT

Prevalent retrieval-based tool-use pipelines struggle with a dual semantic challenge: their retrievers often employ encoders that fail to capture complex semantics, while the Large Language Model (LLM) itself lacks intrinsic tool knowledge from its natural language pretraining. Generative methods offer a powerful alternative by unifying selection and execution, tasking the LLM to directly learn and generate tool identifiers. However, the common practice of mapping each tool to a unique new token introduces substantial limitations: it creates a scalability and generalization crisis, as the vocabulary size explodes and each tool is assigned a semantically isolated token. This approach also creates a semantic bottleneck that hinders the learning of collaborative tool relationships, as the model must infer them from sparse co-occurrences of monolithic tool IDs within a vast library. To address these limitations, we propose **ToolWeaver**, a novel generative tool learning framework that encodes tools into hierarchical sequences. This approach makes vocabulary expansion logarithmic to the number of tools. Crucially, it enables the model to learn collaborative patterns from the dense co-occurrence of shared codes, rather than the sparse co-occurrence of monolithic tool IDs. We generate these structured codes through a novel tokenization process designed to weave together a tool's intrinsic semantics with its extrinsic co-usage patterns. These structured codes are then integrated into the LLM through a generative alignment stage, where the model is fine-tuned to produce the hierarchical code sequences. Evaluation results with nearly 47,000 tools show that **ToolWeaver** significantly outperforms state-of-the-art methods, establishing a more scalable, generalizable, and semantically-aware foundation for advanced tool-augmented agents.[1]

## 1 INTRODUCTION

LLMs have rapidly evolved into powerful interactive agents by integrating with external tools, enabling them to access dynamic information and perform comprehensive real-world tasks (Yao et al., 2023; Qin et al., 2023; Hao et al., 2023; Gao et al., 2024b; Wang et al., 2024b; Zhao et al., 2025; Cheng et al., 2025). Concurrently, the number and diversity of available tools have grown substantially, ranging from general services to domain-specific APIs, leading to significant challenges such as scalability and generalization for tool selection and execution (Mialon et al., 2023).

Existing methods (Patil et al., 2023; Hao et al., 2023; Paranjape et al., 2023; Yao et al., 2023; Wang et al., 2024b; Qin et al., 2024) have focused on equipping LLMs with tool-use capabilities through

---
[\*]Corresponding authors.
[1]Data and code are available at https://github.com/Fwibo/ToolWeaver

retrieval-based or generative approaches. Retrieval-based methods, such as ToolLLM (Qin et al., 2023) and Gorilla (Patil et al., 2023), employ external retrievers to select tools from a large corpus, which are often constrained by the LLM's input length and add pipeline complexity. In contrast, generative methods (Hao et al., 2023; Wang et al., 2024b; Zhu et al., 2025) offer end-to-end simplicity by fine-tuning the LLM to directly generate tool invocations. A common strategy in this paradigm is to map each tool to a unique special token (Liu et al., 2024).

However, this simple "one-token-per-tool" paradigm suffers from two fundamental drawbacks. Firstly, it faces a **critical scalability and generalization challenge**. As illustrated in Figure 1(a), the vocabulary size grows linearly with the number of tools, which hinders generalization as each new tool requires a semantically isolated token. For a model like Llama-3-8B (Dubey et al., 2024) with a vocabulary of 128,256, integrating a large benchmark like ToolBench would require adding nearly 47,000 new tokens. This massive injection of out-of-vocabulary (OOV) tokens leads to significant memory overhead and risks disrupting the model's pretrained linguistic knowledge, causing a catastrophic degradation of its general language capabilities (Wang et al., 2024a). Secondly, it suffers a **semantic bottleneck for complex reasoning**. By flattening tools into isolated, unique tokens, the model struggles to learn collaborative relationships, as it is forced to rely on the statistically sparse co-occurrence of their individual IDs. For instance, consider the query "is it a good day to take my kid to the park?" as illustrated in Figure 1(b). To answer this comprehensively, a model needs to infer the relationship between tools like `Realtime Weather` and `Air Quality`. However, because the joint appearance of any specific tool pair is rare in a vast library, the model might check the weather but fail to consider air quality, thus providing an incomplete or misleading answer.

To address these challenges, we propose ToolWeaver, a framework that fundamentally rethinks tool representation. Instead of flat identifiers, ToolWeaver represents each tool as a compositional sequence of discrete codes. This hierarchical structure, generated via our novel, unsupervised collaborative-aware vector quantization, is not only highly scalable—reducing vocabulary expansion from linear to logarithmic—but more importantly, it is inherently compatible with the autoregressive nature of LLMs and enables the model to learn from a dense collaborative signal.

By jointly modeling a tool's intrinsic function and its extrinsic co-usage patterns, this method encourages functionally related tools to share codes. For instance, `Realtime Weather` (`<T1_1><T2_1>`) and `Air Quality` (`<T1_1><T2_2>`) can share a parent code (`<T1_1>`) that emerges to group tools for a shared context like "outdoor conditions", allowing the model to learn their collaborative nature from the dense co-occurrence of the shared code rather than the sparse co-occurrence of individual tools. Subsequently, these structured codes are integrated into the LLM via a generative alignment stage, training the model to produce the hierarchical code sequences for complex tool invocation. In summary, our main contributions are as follows:

- We propose ToolWeaver, a novel framework that represents tools as compositional codes. A collaborative-aware tokenization process generates these codes, enabling the model to learn robust collaborative patterns from the dense co-occurrence of shared codes, thus overcoming the scalability and semantic bottlenecks of prior methods while enhancing generalization.
- We introduce a multi-stage generative alignment process that effectively aligns the structured tool codes with the LLM's internal knowledge. This fine-tuning teaches the model to natively generate the hierarchical code sequences, enabling both accurate tool selection and complex external tool use.
- Experimental validation on a large benchmark of nearly 47,000 tools demonstrates that ToolWeaver significantly outperforms state-of-the-art methods in complex task completion while substantially mitigating the impact on the LLM's general capabilities.

## 2 RELATED WORK

### 2.1 LARGE LANGUAGE MODELS WITH TOOLS

Equipping LLMs with external tools (e.g., APIs, knowledge bases) enables them to execute complex, interactive tasks. Current methods are broadly categorized as tuning-free or tuning-based. Tuning-free approaches use in-context learning, placing tool descriptions and examples in the prompt to

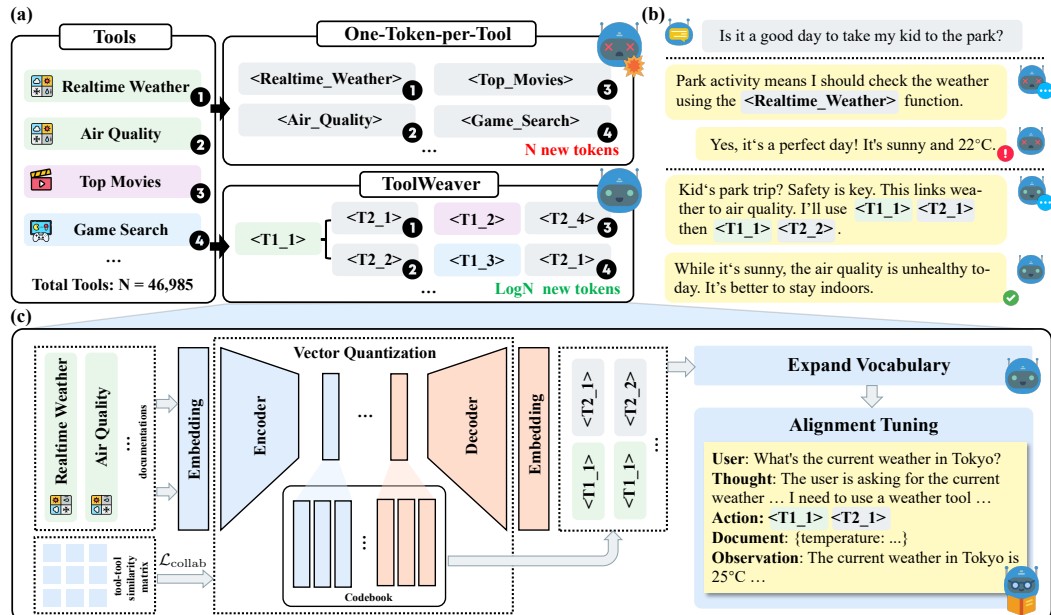

Figure 1: An overview of the ToolWeaver framework. (a) We contrast the standard "one-token-per-tool" method, which creates a massive flat vocabulary, with our compositional approach that scales logarithmically. (b) Our model leverages collaborative signals between tools (e.g., `Realtime Weather` and `Air Quality`) for complex reasoning where "one-token-per-tool" representations fail. (c) The ToolWeaver architecture learns these structured representations through a collaborative-aware vector quantization process, which are then integrated into an LLM.

guide the LLM without updating its parameters (Qin et al., 2024; Paranjape et al., 2023; Yao et al., 2023; Wu et al., 2024; Liu et al., 2025). In contrast, tuning-based methods fine-tune LLMs on tool-use datasets (Schick et al., 2023; Wang et al., 2024b; Su et al., 2025). This has evolved from domain-specific tools like retrieval modules (Gao et al., 2024c) (e.g., WebGPT (Nakano et al., 2022) for web browsing (Brown et al., 2020)) to general-purpose toolsets (Qin et al., 2023; Li et al., 2023) (e.g., Toolformer (Schick et al., 2023) using calculators, QA systems) (Zhuang et al., 2023; Chen et al., 2025).

The growing scale of toolsets, highlighted by benchmarks like ToolBench (Qin et al., 2023) and API-Bank (Li et al., 2023), creates a major scalability problem. In-context methods face context window limitations, while tuning-based methods require constant retraining. This necessitates more efficient tool retrieval and selection mechanisms.

## 2.2 TOOL SELECTION

With expanding toolkits, effective tool selection is critical. One approach is retriever-based selection, which treats tools as documents to be ranked by information retrieval models like BM25 (Robertson & Zaragoza, 2009) or dense retrievers (Karpukhin et al., 2020; Xiong et al., 2020), as used in Gorilla (Patil et al., 2023). While techniques like query rewriting (Chen et al., 2024) and iterative refinement (Xu et al., 2024) improve accuracy, these pipelines suffer from high latency and complexity.

An alternative is generative selection, where each tool is mapped to a single, atomic token that the LLM generates directly (Hao et al., 2023; Wang et al., 2024b). ToolGen (Wang et al., 2024b) exemplifies this by integrating tool knowledge as virtual tokens. However, this approach scales poorly: large tool vocabularies increase memory and latency, artificial tokens can disrupt linguistic priors, and the flat token representation impedes reasoning over semantic tool relationships.

## 2.3 INTEGRATING COLLABORATIVE SEMANTICS INTO LLMS

Recent research seeks to integrate collaborative semantics into LLMs, bridging the semantic gap between their native linguistic space and the symbolic knowledge embedded in collaborative signals (Lin et al., 2025). Existing methods primarily project collaborative knowledge into the LLM's semantic space. For instance, SeLLa-Rec (Wang et al., 2025) maps collaborative knowledge to specialized tokens, while LC-Rec (Zheng et al., 2024) utilizes learning-based vector quantization via a Residual-Quantized Variational AutoEncoder (RQ-VAE) (Lee et al., 2022) to generate structured identifiers. Others leverage graph structures, such as GAL-Rec (Guan et al., 2024), which uses GNN-inspired techniques to teach relational patterns. This principle also extends to generative retrieval, where models like CFRAG (Shi et al., 2025) infuse collaborative filtering into the RAG pipeline.

Despite their progress, these approaches share a fundamental limitation: they all rely on post-hoc alignment. This paradigm introduces semantically isolated identifiers and then forces the model to learn their meaning through a separate alignment phase, disconnected from its foundational representations. This reveals a critical research gap for a more foundational approach that integrates collaborative semantics directly into the tokenization process, enriching representations from the ground up.

## 3 TOOLWEAVER

### 3.1 PRELIMINARIES

Current tool-augmented agents often operate by iteratively reasoning and acting. Given a user query $q$ and a large tool corpus $\mathcal{D} = \{d_1, \ldots, d_N\}$ where $|\mathcal{D}| = N$, an agent typically follows a multi-stage process: (1) planning a step $(p_i)$, (2) selecting a tool $(d_i)$, (3) generating its parameters $(\alpha_i)$, and (4) observing the execution feedback $(f_i)$. This cycle repeats until the task is complete, forming an interaction trajectory $\text{Traj} = [q, (p_1, d_1, \alpha_1, f_1), \ldots, (p_t, d_t, \alpha_t, f_t), a]$. To streamline this process, a promising generative paradigm (Hao et al., 2023; Wang et al., 2024b) reframes tool selection as a next-token prediction task. This is achieved by mapping each tool $d \in \mathcal{D}$ to a unique, specially added token in the language model's vocabulary. While simple, this "one-token-per-tool" scheme suffers from the scalability, generalization, and semantic limitations discussed in the introduction.

To overcome these challenges, we propose ToolWeaver. Instead of a single token, it represents each tool as a compositional sequence of discrete codes. As visualized in Figure 1(c), we employ a set of $L$ codebooks, $\mathcal{C} = \{\mathcal{C}_1, \ldots, \mathcal{C}_L\}$, where each codebook $\mathcal{C}_l$ contains $K$ learnable code vectors. Each tool is then mapped to a unique sequence of $L$ indices, $[\iota_1, \iota_2, \ldots, \iota_L]$. This hierarchical structure yields a representation capacity of up to $K^L$ tools while only requiring the addition of $L \times K$ new tokens to the vocabulary, achieving the logarithmic compression contrasted in Figure 1(a).

While this compositional structure is inherently scalable, the key contribution of ToolWeaver is its novel structured tokenization process. An integral aspect of this process is the explicit use of collaborative signals from tool usage data during codebook learning, which ensures the resulting representations are not only semantically coherent but also aligned with their practical, collaborative functions in downstream tasks.

### 3.2 STRUCTURED TOKENIZATION GUIDED BY COLLABORATIVE SEMANTICS

The core of ToolWeaver is its structured tokenization process (see Figure 1(c)), which transforms each tool's unstructured documentation into a compositional representation. This process is designed to embed both semantic and collaborative relationships directly into the code structure. It involves a multi-stage pipeline of initial semantic encoding, collaborative-aware residual quantization, and conflict mitigation.

**Initial Semantic Representation.** Given the textual documentation for each tool $d \in \mathcal{D}$ (including its name and description), we first leverage a powerful pretrained text encoder to generate a dense semantic embedding. This initial representation, denoted as $e_d \in \mathbb{R}^D$, captures the core functionality of the tool.

$$e_d = \text{Text-Encoder}(\text{Doc}_d), \tag{1}$$

these embeddings $\{e_d\}_{d \in \mathcal{D}}$ serve as the foundational input for our structured tokenization framework.

**Collaborative-Aware Residual Quantization.** To create a compact and hierarchical code structure, we employ an RQ-VAE (Lee et al., 2022), a multi-level vector quantization approach. This method operates without requiring human-annotated labels. Unlike traditional single-layer vector quantization or simple clustering algorithms that offer a flat representation, RQ-VAE sequentially quantizes residual errors, allowing it to achieve a significantly larger expression space with a more compact and manageable vocabulary size. As defined in the preliminaries, we use $L$ codebooks, $\{\mathcal{C}_1, \ldots, \mathcal{C}_L\}$, where each codebook $\mathcal{C}_l$ contains $K$ learnable centroid vectors $\{v_{l,k}\}_{k=1}^{K}$, with $v_{l,k} \in \mathbb{R}^{D'}$. For efficiency, we first reduce the dimensionality of the initial embeddings $e_d$ from $D$ to $D'$ using a linear projection, yielding $z_d$.

The quantization process is recursive. For each tool $d$, the initial residual is set to its projected embedding, $r_{d,1} = z_d$. At each level $l \in \{1, \ldots, L\}$, we find the closest centroid in codebook $\mathcal{C}_l$ for the current residual $r_{d,l}$ and subtract it to compute the residual for the next level:

$$\iota_{d,l} = \underset{k \in \{1, \ldots, K\}}{\arg\min} \|r_{d,l} - v_{l,k}\|_2^2, \tag{2}$$

$$r_{d,l+1} = r_{d,l} - v_{l,\iota_{d,l}}, \tag{3}$$

where $\iota_{d,l}$ is the discrete code index assigned to tool $d$ at level $l$. The final quantized representation is the sum of the selected codebook vectors: $\hat{z}_d = \sum_{l=1}^{L} v_{l,\iota_{d,l}}$.

The standard RQ-VAE is trained to minimize a combination of a reconstruction loss and a quantization loss:

$$\mathcal{L}_{\text{recon}} = \|z_d - \hat{z}_d\|_2^2, \tag{4}$$

$$\mathcal{L}_{\text{quant}} = \sum_{l=1}^{L} \left( \|\text{sg}[r_{d,l}] - v_{l,\iota_{d,l}}\|_2^2 + \beta \|r_{d,l} - \text{sg}[v_{l,\iota_{d,l}}]\|_2^2 \right). $$

where $\text{sg}[\cdot]$ is the stop-gradient operator and $\beta$ is a commitment weight, typically set to 0.25.

To ensure the resulting codes capture not only a tool's intrinsic function but also its extrinsic collaborative patterns, we guide the quantization process using a pre-computed tool-tool similarity matrix $A$.

This matrix is derived from a tool co-occurrence matrix $C$ built from the usage trajectories, where each element $C_{uv}$ counts the total number of times tools $u$ and $v$ appear together. The similarity score $A_{uv}$ is then calculated using cosine similarity:

$$A_{uv} = \frac{C_{uv}}{\sqrt{C_{uu} \cdot C_{vv}}}, \tag{5}$$

where $C_{uu}$ and $C_{vv}$ represent the total occurrence counts for tools $u$ and $v$, respectively.

We introduce a graph Laplacian regularization term that encourages similar tools to have nearby quantized representations:

$$\mathcal{L}_{\text{collab}} = \sum_{u,v \in \mathcal{D}} A_{uv} \|\hat{z}_u - \hat{z}_v\|_2^2. \tag{6}$$

This term penalizes large distances between the quantized representations of tools that frequently co-occur or are functionally related. This integration with RQ is crucial, as its multi-layer, residual nature facilitates a progressive refinement of collaborative semantics: the initial layers capture broad functional similarities, while subsequent layers model finer distinctions on the residual information. Combining these objectives, the final training objective for our structured tokenization becomes:

$$\mathcal{L}_{\text{tokenize}} = \mathbb{E}_{d \sim \mathcal{D}}[\mathcal{L}_{\text{recon}} + \mathcal{L}_{\text{quant}}] + \lambda \mathcal{L}_{\text{collab}}, \tag{7}$$

where $\lambda$ is a hyperparameter balancing the reconstruction fidelity with the collaborative structure alignment.

**Conflict Mitigation via Uniform Mapping.** A practical challenge in any tree-based or multi-level quantization is index collision, where multiple distinct tools map to the exact same sequence of code indices $[\iota_1, \ldots, \iota_L]$. A naive solution of adding an extra, semantically meaningless layer of IDs is undesirable, as it can disrupt the learned semantic structure. To resolve this while preserving semantic integrity, we enforce a uniform mapping constraint on the final codebook $\mathcal{C}_L$.

Our objective is to ensure that tool representations are distributed as uniformly as possible across the centroids of the final codebook. We formulate this as a constrained optimization problem, adapting the standard quantization objective for the final level $L$. For a batch of final-level residuals $\mathcal{B}_L = \{r_{d,L}\}_{d \in \text{batch}}$, we aim to solve:

$$\min_{\Pi} \sum_{d \in \mathcal{B}_L} \sum_{k=1}^{K} \pi_{dk} \|r_{d,L} - v_{L,k}\|_2^2, \tag{8}$$

subject to:

$$\forall d, \quad \sum_{k=1}^{K} \pi_{dk} = 1, \forall k, \quad \sum_{d \in \mathcal{B}_L} \pi_{dk} = \frac{|\mathcal{B}_L|}{K}, \tag{9}$$

where $\pi_{dk} = p(\iota_{d,L} = k | r_{d,L})$ represents the soft assignment probability of tool $d$'s residual to centroid $k$. The first constraint ensures that each tool's residual is fully assigned across the codebook. The second, more critical constraint, enforces that each centroid in the final codebook is assigned an equal share of tools from the batch, thereby mitigating collisions.

Following Zheng et al. (2024), we frame this as an optimal transport problem where $\|r_{d,L} - v_{L,k}\|_2^2$ is the transport cost. This formulation allows us to find an optimal assignment matrix $\Pi$ that satisfies the uniform distribution constraint. In our implementation, we solve this problem efficiently using the Sinkhorn-Knopp algorithm (Cuturi, 2013). This strategy promotes a unique identifier for every tool without compromising the learned semantic space.

### 3.3 Multi-step Generative Alignment Tuning

The final stage of our framework is to integrate these structured codes into the LLM via generative alignment tuning. The code sequence for each tool, e.g., $[\iota_{d,1}, \ldots, \iota_{d,L}]$, is mapped to a sequence of new, unique tokens (e.g., `<T1_1><T2_1>`) added to the LLM's vocabulary. Let $\iota_d$ denote this sequence of code-tokens. The corresponding embeddings for these new vocabulary tokens are randomly initialized. We then fine-tune the model in two stages:

**Stage 1: Tool Retrieval Alignment.** The model learns to generate the correct tool's code sequence $\iota_d$ from a user query $q$ by fine-tuning on a dataset of query-tool pairs.

$$\mathcal{L}_{\text{retrieval}} = -\mathbb{E}_{(q,d)}[\log P(\iota_d | q)]. \tag{10}$$

**Stage 2: Tool Usage Trajectory Alignment.** We further fine-tune the model on full interaction trajectories. The model learns to generate sequences of reasoning, actions (tool calls, including their code-tokens $\iota_d$ and parameters $\alpha_d$), and final answers, with the loss computed only over the assistant's tokens.

This progressive tuning aligns the model for both accurate tool selection and effective execution in downstream tasks.

### 3.4 Inference

To prevent the model from generating invalid tool codes during inference, we employ a constrained beam search, a standard technique in similar generative frameworks (Wang et al., 2024b). A pre-computed prefix tree (trie) of all valid tool code sequences ($\iota_d$ for all $d \in \mathcal{D}$) guides the search, ensuring that only valid identifiers are generated by masking the logits of invalid next tokens at each step. This constraint is applied only during the tool selection phase, preserving the model's full generative capacity for other tasks.

Table 1: Tool retrieval evaluation performance on ToolBench. Performance is measured by NDCG@k across varying query complexities (I1-I3). ToolWeaver consistently outperforms both retrieval-based (BM25, EmbSim, ToolRetriever) and generative (ToolGen) methods. * represents the results disclosed in Wang et al. (2024b), while the others are the results we re-implemented based on the open-source checkpoints.

| Model | I1 | | | I2 | | | I3 | | |
|---|---|---|---|---|---|---|---|---|---|
| | NDCG@1 | NDCG@3 | NDCG@5 | NDCG@1 | NDCG@3 | NDCG@5 | NDCG@1 | NDCG@3 | NDCG@5 |
| BM25* | 22.77 | 22.64 | 25.61 | 18.29 | 20.74 | 22.18 | 10.00 | 10.08 | 12.33 |
| EmbSim* | 54.00 | 50.82 | 55.86 | 40.84 | 36.67 | 39.55 | 18.00 | 17.77 | 20.70 |
| ToolRetriever* | 72.31 | 70.30 | 74.99 | 64.54 | 57.91 | 63.61 | 52.00 | 39.89 | 42.92 |
| ToolGen* | 87.67 | 88.84 | 91.54 | 83.46 | 86.24 | 88.84 | 79.00 | 79.80 | 84.79 |
| BM25 | 26.92 | 26.13 | 29.00 | 20.00 | 21.92 | 23.46 | 10.00 | 10.08 | 12.33 |
| EmbSim | 50.50 | 48.15 | 53.41 | 46.00 | 39.58 | 43.05 | 18.00 | 17.77 | 20.94 |
| ToolRetriever | 75.92 | 76.96 | 82.31 | 63.00 | 66.38 | 72.72 | 28.00 | 39.28 | 44.54 |
| ToolGen | 88.50 | 88.83 | 91.65 | 84.00 | 85.65 | 89.02 | 81.00 | 80.83 | 85.83 |
| ToolWeaver | **91.16** | **91.14** | **93.48** | **89.76** | **89.70** | **91.80** | **88.00** | **85.80** | **90.12** |

# 4 EXPERIMENTS

We conduct extensive experiments to evaluate ToolWeaver on large-scale tool retrieval and end-to-end evaluation, focusing on performance, generalization, and scalability against state-of-the-art methods.

## 4.1 EXPERIMENTAL SETUP

**Dataset.** We use the large-scale ToolBench benchmark (Qin et al., 2023), which consists of over 16,000 tool collections comprising 46,985 unique APIs. Although a tool collection may contain multiple APIs, for simplicity, we refer to each individual API as a "tool" in this paper.

The dataset's structure allows evaluation across scenarios of increasing complexity, from simple single-tool tasks (I1), to multi-tool planning within a single category (I2), and finally to complex orchestration of tools across different categories (I3) (Qin et al., 2023). Furthermore, to rigorously assess generalization, we adopt fine-grained splits: I1 Tool., I1 Cat., and I2 Cat., where "Tool." and "Cat." denote tools and categories, respectively, that are unseen during training. All data for our retrieval and agent-tuning experiments are converted from this benchmark, with further details in Appendix A.1.

**Baselines.** We compare ToolWeaver against a comprehensive set of baselines for both tool retrieval and end-to-end task completion. For retrieval evaluation, we use the classic unsupervised methods BM25 (Robertson & Zaragoza, 2009) and Embedding Similarity (EmbSim), alongside the state-of-the-art supervised models ToolRetriever (Qin et al., 2023) and ToolGen (Wang et al., 2024b). For end-to-end evaluation, we benchmark against strong generative models including GPT-4o-mini, ToolLlama-2 (Qin et al., 2023), and ToolGen (Wang et al., 2024b). A detailed description of each baseline is provided in Appendix A.2.

**Metrics.** For tool retrieval, we use Normalized Discounted Cumulative Gain (NDCG@k) (Järvelin & Kekäläinen, 2002) for k={1,3,5}, which evaluates the ranking quality of retrieved tools by considering both relevance and position. For the agent task, we adopt the StableToolBench framework (Guo et al., 2024) and report two key metrics: Solvable Pass Rate (SoPR), the percentage of tasks successfully completed, and Solvable Win Rate (SoWR), which measures the quality of the final answer against a strong reference model.

**Implementation Details.** For our main experiments presented in the body of this paper, we use the pretrained Llama-3-8B as the primary foundation model for both ToolWeaver and key generative baselines to ensure a fair comparison. To demonstrate the robustness and generalizability of our approach across different architectures, we provide a full set of supplementary results using the Qwen model series (Yang et al., 2025) in Appendix B.3. All other architectural choices, training procedures, and hyperparameter settings are detailed in Appendix A.3.

## 4.2 RESULTS

Table 1 presents a comprehensive comparison of tool retrieval evaluation performance. Across all query complexities (I1-I3), ToolWeaver consistently outperforms all baselines. In the most complex I3 scenario, ToolWeaver achieves an NDCG@1 of 88.00, significantly higher than ToolGen (81.00) and all retrieval-based methods. Table 2 details the end-to-end evaluation results. Tool-

Table 2: Comparison of end-to-end evaluation performance on ToolBench, measured by Solvable Pass Rate (SoPR) and Solvable Win Rate (SoWR). The SoWR is calculated against the GPT-4o-mini baseline. GPT-4o-mini and ToolLlama-2 are tested in a challenging Retrieval setting (Re.) that requires selecting tools from the full set. In contrast, ToolGen and ToolWeaver generate tool tokens directly, without the need for a retriever. ToolWeaver outperforms other models in diverse scenarios, highlighting its effectiveness in both tool selection and execution.

| Model | Set. | SoPR | | | | | | SoWR | | | | | |
| | | I1 | I2 | I3 | I1-Tool. | I1-Cat. | I2-Cat. | I1 | I2 | I3 | I1-Tool. | I1-Cat. | I2-Cat. |
| --- | --- | --- | --- | --- | --- | --- | --- | --- | --- | --- | --- | --- | --- |
| GPT-4o-mini | Re. | 52.25 | 40.41 | 24.86 | 53.16 | 50.11 | 39.38 | - | - | - | - | - | - |
| ToolLlama-2 | Re. | 28.94 | 24.69 | 10.93 | 28.48 | 36.93 | 19.09 | 25.15 | 30.19 | 24.59 | 26.58 | 27.45 | 20.16 |
| ToolGen | | 52.97 | **45.13** | 36.34 | 45.36 | 55.56 | 45.56 | 36.20 | 42.45 | 49.18 | 32.91 | 42.48 | **37.90** |
| **ToolWeaver** | | **53.17** | 44.03 | **52.19** | **54.85** | **57.41** | **46.24** | **40.49** | **48.11** | **59.02** | **36.08** | **43.14** | 35.48 |

Weaver achieves superior performance in most cases, excelling in both task completion (SoPR) and solution quality (SoWR). In the challenging retrieval setting, ToolWeaver attains the highest SoPR scores in nearly all scenarios. Its advantages are clear not only in simple (I1) and complex multi-tool tasks (I3), but also in generalizing to both unseen tools (I1-Tool) and unseen categories (I1-Cat). The substantial lead in the multi-tool I3 scenario (52.19 vs. ToolGen's 36.34) particularly underscores the effectiveness of our collaborative-aware design for complex planning. Regarding SoWR against the GPT-4o-mini reference, ToolWeaver again demonstrates superior performance in the majority of scenarios. The advantage is particularly pronounced in the most complex I3 tasks, where it achieves a win rate of 59.02, substantially outperforming ToolGen (49.18). Full results across all settings, including additional baselines and evaluation domains, are detailed in Appendix B.1.

## 4.3 ABLATION STUDIES AND ANALYSIS

### 4.3.1 ANALYSIS OF COLLABORATIVE REGULARIZATION WEIGHT

We conducted a sensitivity analysis to assess the impact of the collaborative regularization weight, $\lambda$, on tool selection performance. As shown in Figure 2, model performance across all NDCG metrics consistently improves as $\lambda$ increases from 0.01 to 1, peaking at $\lambda$=1. This trend demonstrates that incorporating collaborative signals is crucial for learning a semantically rich representation that captures how tools function together. However, when $\lambda$ is increased further to 10, performance declines, indicating that an excessively strong collaborative prior can distort tool representations by sacrificing the fidelity of their intrinsic functions. This result empirically validates our central hypothesis: optimal performance is achieved by striking a balance between a tool's intrinsic semantics and its extrinsic collaborative patterns, confirming the effectiveness of ToolWeaver's design.

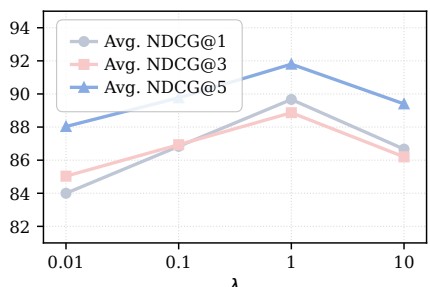

Figure 2: Analysis of the collaborative regularization weight $\lambda$. Performance, measured by average NDCG@k across all I1-I3 scenarios, consistently peaks at $\lambda = 1$.

### 4.3.2 COMPONENT ABLATION ANALYSIS

To demonstrate the cumulative impact of our core components, we performed a step-wise ablation. The results in Figure 3 reveal a clear hierarchy of contributions.

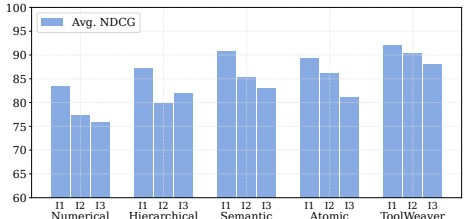
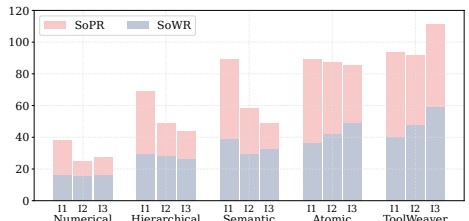

(a) Retrieval Evaluation (Avg. NDCG@{1,3,5}).   (b) End-to-end Evaluation (SoPR/SoWR).

Figure 4: Comparison of tokenization strategies.

The model without semantic initialization (blue bars), which lacks both of our proposed components, performs poorly and serves as a baseline. The first crucial step is adding semantic initialization. This step alone (transitioning from blue to pink bars) causes a dramatic performance leap of over 20 NDCG points, underscoring that a meaningful tool representation is the single most critical foundation.

Building upon this semantically-aware foundation, the final step is to incorporate collaborative guidance (transitioning from pink to grey bars). This yields a further, significant improvement whose magnitude scales with task complexity. The benefit is modest for simple I1 tasks but becomes most pronounced for complex, multi-tool I3 tasks.

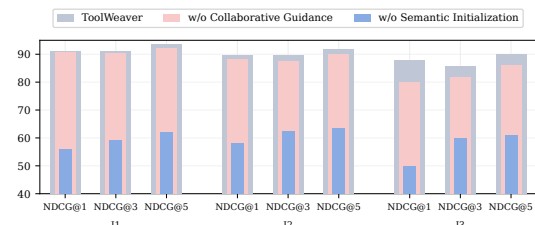

Figure 3: Cumulative ablation analysis of ToolWeaver's components on tool selection (NDCG@k). Performance is shown for the baseline (w/o Semantic Initialization), after adding semantic initialization (w/o Collaborative Guidance), and for the full model.

This trend provides strong evidence for our thesis: while semantic understanding is foundational, explicitly encoding collaborative patterns is the key to mastering complex tool orchestration.

### 4.3.3 COMPARISON OF TOKENIZATION STRATEGIES

To further validate the effectiveness of our collaborative-aware structured tokenization, particularly against other methods that attempt to embed structure or semantics into tool representations, we compare ToolWeaver against several alternative strategies. These baselines represent different paradigms: **Atomic** (as in ToolGen) assigns a single unique token per tool, serving as a flat generative baseline; **Numerical** uses fixed-length numeric strings, providing a minimal-vocabulary but non-semantic baseline; **Hierarchical** generates structured code sequences based on clustering, representing a static, tree-like approach to tokenization; and **Semantic** leverages human-readable parts of tool names, reflecting a direct, lexicon-based semantic approach. Detailed descriptions of these methods are provided in Appendix B.2.

As shown in Figure 4, ToolWeaver consistently outperforms all other tokenization strategies across both tool retrieval and end-to-end evaluation. Notably, while the Hierarchical and Semantic methods attempt to incorporate structure, they struggle to outperform the strong Atomic baseline, particularly in the end-to-end evaluation (SoPR/SoWR). This indicates that simply adding a naive structure is not sufficient for improving performance.

These results underscore the core advantage of ToolWeaver: it does not merely rely on pre-existing hierarchies or basic semantic similarity. Instead, it uniquely weaves collaborative signals into a structured, semantic representation. The underperformance of other structured methods compared to the Atomic baseline highlights that the quality and type of encoded information are critical. ToolWeaver's holistic approach creates tool codes that are not only descriptive of a tool's function but also predictive of its role in complex, multi-tool workflows, leading to superior performance.

### 4.3.4 Impact on General Language Capabilities

A critical concern with generative tool learning is that adding new tokens might disrupt the pre-trained linguistic knowledge of the LLM. Methods like ToolGen inject nearly 47,000 new tokens, linearly expanding the vocabulary and potentially skewing the model's internal distribution. In contrast, ToolWeaver employs a logarithmic expansion strategy that adds significantly fewer tokens.

To rigorously quantify this impact, we selected two complementary evaluation protocols. First, we measured Perplexity (PPL) on WikiText-2 (Merity et al., 2016) to assess how well the model maintains the original probability distribution of natural language. Second, we evaluated Text Summarization on CNN/DailyMail (See et al., 2017) and XSum (Narayan et al., 2018) to verify the model's ability to generate coherent and high-quality text in zero-shot settings.

Table 3: Impact of tool vocabulary expansion on general language capabilities. We report Perplexity (lower is better) and Summarization BERTScore F1 (higher is better). ToolWeaver preserves the base model's capabilities significantly better than ToolGen.

| Model | Language Modeling (PPL) | Text Summarization (BERTScore F1) | | |
|---|---|---|---|---|
| | WikiText-2 (↓) | CNN/DM (↑) | XSum (↑) | Avg. Drop |
| Llama-3-8B (Base) | 6.34 | 85.35 | 85.05 | - |
| ToolGen | 104.54 | 82.93 | 82.53 | 2.47 |
| **ToolWeaver** | **25.36** | **85.07** | **84.18** | **0.57** |

Table 3 summarizes the results. The linear vocabulary expansion in ToolGen leads to catastrophic degradation. Specifically, its PPL on WikiText-2 increases drastically to 104.54, which is approximately 16 times that of the base model. Furthermore, its generation quality drops notably on the abstractive XSum benchmark. In contrast, ToolWeaver demonstrates superior robustness. The PPL remains much lower at 25.36, and the summarization performance on CNN/DailyMail is nearly identical to the base model, achieving a BERTScore of 85.07 compared to the original 85.35.

These findings indicate that our structured tokenization preserves the linguistic core of the LLM far better than assigning isolated atomic tokens to a vast tool library. We provide additional evaluations on general understanding benchmarks, such as MMLU, along with detailed experimental configurations in Appendix B.4.

## 5 Conclusion

In this paper, we introduced ToolWeaver, a framework designed to address the critical scalability, generalization, and semantic challenges of the "one-token-per-tool" paradigm. ToolWeaver represents each tool as a hierarchical sequence of discrete codes, making vocabulary expansion logarithmic to the number of tools. Through a novel, collaborative-aware tokenization process, our framework weaves a tool's intrinsic semantics with its extrinsic co-usage patterns, encouraging functionally related tools to share codes. This allows the model to learn robust collaborative patterns from the dense co-occurrence of shared codes, rather than the sparse co-occurrence of isolated tool IDs. Extensive experiments on the ToolBench benchmark demonstrate that ToolWeaver significantly outperforms state-of-the-art methods in complex task completion and generalization, while better preserving the model's general language capabilities. Our work establishes a more scalable, generalizable, and semantically-aware foundation for building advanced tool-using agents, with future directions including reinforcement learning to autonomously discover collaborative patterns.

## 6 Acknowledgments

This work is supported by the Zhongguancun Academy (Grant No. 2608050301, Information Intelligence Project), the Beijing Natural Science Foundation (Grant No. L252033), and the National Natural Science Foundation of China (Grant Nos. 62236010, 62141608, 62206291, 62372454).

ETHICS STATEMENT

Our work aims to advance the capabilities of tool-using AI agents. We acknowledge the potential for misuse, as more capable agents could be directed to interact with malicious APIs. Furthermore, the ToolBench dataset, while based on real-world tools, was not audited for biases or privacy risks. We present this as foundational research and emphasize that any real-world deployment requires robust safety protocols and human oversight.

REPRODUCIBILITY STATEMENT

To ensure reproducibility, our complete source code is provided in the supplementary material. All experimental configurations, including dataset processing (Appendix A.1), baseline details (Appendix A.2), and implementation hyperparameters (Appendix A.3), are thoroughly documented. Our experiments utilize the public ToolBench dataset and the StableToolBench evaluation framework.

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

## APPENDIX

## A  EXPERIMENTAL SETUP DETAILS

### A.1  DATASET DETAILS

Our experiments are conducted on the **ToolBench** benchmark (Qin et al., 2023), a large-scale and comprehensive dataset designed for evaluating tool-using agents. ToolBench is constructed upon a vast collection of real-world, high-quality REST APIs sourced from RapidAPI, a major API hub. This grounding in real-world services ensures that the tasks and tools reflect practical challenges faced by AI agents.

**Overall Statistics.**  The full dataset encompasses 46,985 tools (APIs) organized into 129 tool collections. As mentioned in the main text, each tool is annotated with rich metadata, including a human-assigned functional category. There are 49 distinct functional categories in total (e.g., Finance, Travel, Sports, etc.), which provide a semantic grouping for the tools.

**Evaluation Scenarios.**  ToolBench defines a standardized test set comprising 641 queries, which are categorized into three levels of increasing difficulty based on the complexity of the required tool interactions. These scenarios are:

- **Instruction 1 (I1): Single-Tool Usage.** These tasks require the agent to select and correctly use a single tool to answer the user's query. This scenario primarily tests the agent's ability for accurate tool retrieval and basic API execution.

- **Instruction 2 (I2): Intra-Category Multi-Tool Usage.** These tasks involve solving a problem that requires composing a sequence of tools. Critically, all necessary tools belong to the *same functional category*. This tests the agent's ability to reason and plan within a coherent semantic domain.

- **Instruction 3 (I3): Intra-Collection Multi-Tool Usage.** This is the most challenging scenario. Tasks require the agent to orchestrate multiple tools that may come from *different functional categories* but are part of the same broader tool collection (e.g., a "Trip Planning" collection might contain tools from "Flights", "Hotels", and "Maps" categories). This evaluates the agent's advanced planning and generalization capabilities across diverse tool functions.

**Data Statistics for Alignment Tuning Stages.** Our training methodology is structured into two main fine-tuning stages. We utilize the official splits and data provided by the ToolBench benchmark (Qin et al., 2023), processing them to fit our generative framework. The statistics for each stage are detailed below:

- **Stage 1: Tool Retrieval Alignment.** The initial fine-tuning stage is designed to teach the model the crucial mapping between a user's intent and the appropriate tool. To achieve this, we fine-tune the model on Query-Tool pairs extracted from ToolBench. In this supervised task, the input is a natural language query, and the target output is the corresponding tool's structured semantic code sequence. Following the data processing approach of prior work (Wang et al., 2024b), we utilize a comprehensive set of 489,702 query-tool pairs, aggregated across the I1, I2, and I3 scenarios, to train a robust retrieval capability.

- **Stage 2: Tool Usage Trajectory Alignment.** After the model has learned to retrieve tools, the second stage trains it to function as a complete, autonomous agent. This is accomplished by fine-tuning on full execution trajectories. Each trajectory provides a complete, multi-step example of how to reason, plan, generate arguments, and invoke tools to solve a complex user query. We adapt the original ToolBench trajectories by replacing all tool names with our learned semantic codes. For this final and most complex training step, we use a total of 183,336 trajectories.

## A.2 BASELINE DETAILS

In our experiments, we compare ToolWeaver against several representative retrieval and tool-use models. These baselines are chosen to cover a wide range of approaches, from classic unsupervised methods to state-of-the-art generative agents.

- **BM25** (Robertson & Zaragoza, 2009): An unsupervised, classical retrieval model that ranks documents based on query relevance, using normalized term frequency and document length. It serves as a strong lexical baseline.

- **Embedding Similarity (EmbSim)**: An unsupervised semantic retrieval method. We use OpenAI's powerful `text-embedding-3-large` model to generate embeddings for both queries and tool documents, and rank tools based on the cosine similarity of their embeddings.

- **ToolRetriever** (Qin et al., 2023): A supervised, BERT-based dense retriever specifically designed for tool retrieval. It is trained using contrastive learning to distinguish between relevant and irrelevant tools by maximizing the similarity between queries and their corresponding ground-truth tools.

- **ToolGen** (Wang et al., 2024b): A state-of-the-art generative model that unifies tool retrieval and calling. It represents each tool as a unique atomic token and is fine-tuned to directly generate the tool's token and its arguments in response to a query.

- **ToolLlama-2** (Qin et al., 2023): A version of the Llama-2 model fine-tuned for tool use. Unlike generative models like ToolGen and ToolWeaver, it relies on an external retriever to

first select a set of candidate tools, which are then provided in the prompt context for the model to perform reasoning and task completion.

- **GPT-4o-mini**: A highly capable and efficient model from OpenAI. We use it as a strong baseline for end-to-end task completion. Following the StableToolBench evaluation protocol (Guo et al., 2024), it also serves as the reference model for calculating the Solvable Win Rate (SoWR) metric.

- **Re-Invoke** (Chen et al., 2024): An advanced unsupervised retrieval method that enriches tool documents by generating synthetic queries. During inference, it uses an LLM to analyze user intent and employs a multi-view similarity ranking strategy to identify the most relevant tools.

- **IterFeedback** (Xu et al., 2024): A retrieval framework that enhances a BERT-based retriever by incorporating iterative feedback from a large language model. The LLM is prompted to analyze initial retrieval results and provide feedback to refine the search, improving retrieval accuracy over multiple steps.

## A.3 IMPLEMENTATION AND TRAINING DETAILS

This appendix provides a detailed description of the implementation specifics for the models and experiments presented in the main paper, ensuring full reproducibility.

The structured tokenization process begins with generating initial dense semantic embeddings for each tool. We process the textual documentation of each tool (including its name and description) using the `all-mpnet-base-v2` model from the Sentence-Transformers library, which produces a 768-dimensional embedding. The core of our structured tokenization is a collaborative-aware residual quantization process. This process employs a multi-level scheme with $L = 2$ codebooks, $\mathcal{C}_1$ and $\mathcal{C}_2$, each containing $K = 1024$ learnable vectors. This compositional structure represents the entire tool library with only $2 \times 1024 = 2048$ new tokens added to the LLM's vocabulary. The initial 768-dimensional embeddings are first projected into a 64-dimensional space ($D' = 64$) using a multi-layer perceptron (MLP) with sequential hidden layer dimensions of 1024, 512, 256, and 128.

The codebooks are trained using the AdamW optimizer with a learning rate of 1e-5 and a batch size of 5096, over 50 warmup epochs with no weight decay. For the main results, the collaborative regularization weight, $\lambda$, was set to 1.0. We initialize the codebook centroids using k-means with a maximum of 100 iterations. For conflict mitigation, the Sinkhorn-Knopp algorithm is run for 50 iterations.

The integration of these learned codes into the LLM is achieved through a two-stage generative alignment process. In Stage 1, the model is fine-tuned for 5 epochs on query-tool pairs for retrieval alignment. In Stage 2, it is fine-tuned for an additional 2 epochs on full interaction trajectories to learn complex planning. For both stages, we employ a cosine learning rate scheduler with a 3% warmup ratio and a peak learning rate of $4 \times 10^{-5}$. The input context length is truncated to 6,144 tokens. All experiments were conducted on NVIDIA A100 GPUs, and we leveraged the DeepSpeed ZeRO-3 optimization suite and FlashAttention-2 to enhance training efficiency.

## A.4 EVALUATION SETTING DETAILS

Our experiments adopt two distinct retrieval settings from prior work (Wang et al., 2024b): **In-Domain** and **Multi-Domain**. The In-Domain setting restricts the search space to a pre-filtered tool category, while the more challenging Multi-Domain setting requires the model to select from the entire global corpus of nearly 47,000 APIs for any given query. For our primary experiments presented in the main body of this paper, we focus on the Multi-Domain setting as it provides a more realistic and rigorous test of a model's ability to handle large-scale retrieval and disambiguate tool functions. A complete set of results for both settings, including the In-Domain evaluation, is provided for reference in Appendix B.1.

Table 4: Tool retrieval evaluation across two settings: In-domain and Multi-domain. * represents the results disclosed in Wang et al. (2024b), while the others are the results we re-implemented based on the open-source checkpoints.

| Model | I1 | | | I2 | | | I3 | | |
|---|---|---|---|---|---|---|---|---|---|
| | NDCG@1 | NDCG@3 | NDCG@5 | NDCG@1 | NDCG@3 | NDCG@5 | NDCG@1 | NDCG@3 | NDCG@5 |
| **In-domain** | | | | | | | | | |
| BM25* | 29.46 | 31.12 | 33.27 | 24.13 | 25.29 | 27.65 | 32.00 | 25.88 | 29.78 |
| EmbSim* | 63.67 | 61.03 | 65.37 | 49.11 | 42.27 | 46.56 | 53.00 | 46.40 | 52.73 |
| Re-Invoke* | 69.47 | - | 61.10 | 54.56 | - | 53.79 | 59.65 | - | 59.55 |
| IterFeedback* | 90.70 | 90.95 | 92.47 | 89.01 | 85.46 | 87.10 | **91.74** | **87.94** | 90.20 |
| ToolRetriever* | 80.50 | 79.55 | 84.39 | 71.18 | 64.81 | 70.35 | 70.00 | 60.44 | 64.70 |
| ToolGen* | 89.17 | 90.85 | 92.67 | 91.45 | 88.79 | 91.13 | 87.00 | 85.59 | 90.16 |
| BM25 | 29.25 | 31.04 | 33.49 | 26.50 | 25.97 | 27.96 | 32.00 | 25.88 | 29.78 |
| EmbSim | 61.00 | 57.78 | 62.31 | 54.00 | 45.31 | 49.54 | 54.00 | 46.56 | 52.91 |
| ToolRetriever | 83.50 | 83.67 | 88.66 | 72.00 | 73.27 | 80.40 | 70.00 | 70.01 | 77.21 |
| ToolGen | 91.00 | 92.15 | 94.11 | 87.50 | 88.52 | 90.81 | 87.00 | 85.35 | 90.08 |
| ToolWeaver | **93.76** | **94.80** | **95.69** | **91.91** | **93.07** | **95.63** | 86.00 | 86.13 | **90.39** |
| **Multi-domain** | | | | | | | | | |
| BM25* | 22.77 | 22.64 | 25.61 | 18.29 | 20.74 | 22.18 | 10.00 | 10.08 | 12.33 |
| EmbSim* | 54.00 | 50.82 | 55.86 | 40.84 | 36.67 | 39.55 | 18.00 | 17.77 | 20.70 |
| ToolRetriever* | 72.31 | 70.30 | 74.99 | 64.54 | 57.91 | 63.61 | 52.00 | 39.89 | 42.92 |
| ToolGen* | 87.67 | 88.84 | 91.54 | 83.46 | 86.24 | 88.84 | 79.00 | 79.80 | 84.79 |
| BM25 | 26.92 | 26.13 | 29.00 | 20.00 | 21.92 | 23.46 | 10.00 | 10.08 | 12.33 |
| EmbSim | 50.50 | 48.15 | 53.41 | 46.00 | 39.58 | 43.05 | 18.00 | 17.77 | 20.94 |
| ToolRetriever | 75.92 | 76.96 | 82.31 | 63.00 | 66.38 | 72.72 | 28.00 | 39.28 | 44.54 |
| ToolGen | 88.50 | 88.83 | 91.65 | 84.00 | 85.65 | 89.02 | 81.00 | 80.83 | 85.83 |
| ToolWeaver | **91.16** | **91.14** | **93.48** | **89.76** | **89.70** | **91.80** | **88.00** | **85.80** | **90.12** |

## B SUPPLEMENTARY EXPERIMENTAL RESULTS

### B.1 EXTENDED RESULTS ON MAIN EXPERIMENTS

This section provides a more detailed and comprehensive view of our experimental results, supplementing the key findings presented in the main paper. While the main text focused on the most challenging Multi-Domain setting to rigorously evaluate ToolWeaver's performance in a realistic, large-scale environment, we present results for both In-Domain and Multi-Domain settings here for completeness and to facilitate a thorough comparison with prior work.

Tables 4 offers a complete breakdown of the tool retrieval evaluation. We include results reported by the original authors of baseline methods (*) alongside our own reproductions. The strong alignment between our re-implemented results and those originally published for models like ToolGen validates the fairness and correctness of our experimental setup. Even in the In-Domain setting, where the search space is constrained, ToolWeaver demonstrates top-tier performance. It is particularly noteworthy that ToolWeaver, as a single end-to-end model, outperforms complex, multi-stage retrieval systems like IterFeedback in most scenarios, highlighting the efficiency of our generative approach.

In Table 5, we expand on the end-to-end task completion evaluation. For full transparency, this table includes results from prior work (*) alongside our own. It is important to note potential differences in evaluation protocols. For example, some prior results were obtained using GPT-3.5 as the core agent and evaluator. Considering that GPT-3.5 is no longer a state-of-the-art model and its usage can be costly, we chose to standardize our evaluation using the more recent and capable GPT-4o-mini as both a strong baseline and, for SoWR, the reference judge. This ensures a consistent and modern benchmark for all models we tested. Despite these variations, ToolWeaver consistently demonstrates superior or highly competitive performance. Its significant lead in complex multi-step tasks (I3) and generalization scenarios remains evident, underscoring the benefits of its collaborative-aware tokenization for robust task planning and execution. We also include the SoWR results for GPT-4o-mini in this table for completeness; however, similar to observations in other studies, we noted a tendency for the model to favor its own solutions, which is why we focused on comparing the fine-tuned models in the main text to ensure a fair assessment.

### B.2 DETAILED COMPARISON OF TOKENIZATION STRATEGIES

To provide a comprehensive evaluation of our tokenization strategy, we implemented and compared it against several representative baseline methods. This section describes these alternatives. For all

Table 5: Tool calling evaluation performance on unseen instructions and unseen tools under two settings. Bold values denote the highest performance, considering only the results reproduced in our experimental setting.

| Model | Setting | SoPR | | | | | | SoWR | | | | | |
|---|---|---|---|---|---|---|---|---|---|---|---|---|---|
| | | I1 | I2 | I3 | I1-Tool. | I1-Cat. | I2-Cat. | I1 | I2 | I3 | I1-Tool. | I1-Cat. | I2-Cat. |
| GPT-3.5* | Retrieval | 51.43 | 41.19 | 34.43 | 57.59 | 53.05 | 46.51 | 53.37 | 53.77 | 37.70 | 46.20 | 54.25 | 54.81 |
| ToolLlama-2* | Retrieval | 56.13 | 49.21 | 34.70 | - | - | - | 50.92 | 53.77 | 21.31 | - | - | - |
| ToolLlama* | Retrieval | 54.60 | 49.96 | 51.37 | 57.70 | 61.76 | 45.43 | 49.08 | 61.32 | 31.15 | 48.73 | 50.98 | 44.35 |
| ToolGen* | | 56.13 | 52.20 | 47.54 | 56.54 | 49.46 | 51.96 | 50.92 | 62.26 | 34.42 | 40.51 | 39.87 | 37.90 |
| GPT-4o-mini | Retrieval | 52.25 | 40.41 | 24.86 | 53.16 | 50.11 | 39.38 | **47.24** | **52.83** | 44.26 | **49.37** | **50.33** | **42.74** |
| ToolLlama-2 | Retrieval | 28.94 | 24.69 | 10.93 | 28.48 | 36.93 | 19.09 | 25.15 | 30.19 | 24.59 | 26.58 | 27.45 | 20.16 |
| ToolGen | | 52.97 | **45.13** | 36.34 | 45.36 | 55.56 | 45.56 | 36.20 | 42.45 | 49.18 | 32.91 | 42.48 | 37.90 |
| **ToolWeaver** | | **53.17** | 44.03 | **52.19** | **54.85** | **57.41** | **46.24** | 40.49 | 48.11 | **59.02** | 36.08 | 43.14 | 35.48 |

methods, we follow the same two-stage generative alignment tuning process described in Section 3.3 to ensure a fair comparison of the representation strategies themselves.

The detailed performance results for tool retrieval and end-to-end task completion are presented in Table 6 and Table 7, respectively. These tables provide the underlying data for the summary chart in Figure 4 of the main paper.

Table 6: Retrieval performance (NDCG@k) of different tokenization methods. ToolWeaver's approach of integrating collaborative semantics into a structured representation yields the best performance, especially in complex multi-tool scenarios (I2, I3).

| Tokenization | I1 | | | I2 | | | I3 | | |
|---|---|---|---|---|---|---|---|---|---|
| | NDCG@1 | NDCG@3 | NDCG@5 | NDCG@1 | NDCG@3 | NDCG@5 | NDCG@1 | NDCG@3 | NDCG@5 |
| Numerical | 81.55 | 83.61 | 85.13 | 76.93 | 77.02 | 78.29 | 71.88 | 75.94 | 79.45 |
| Hierarchical | 86.72 | 85.93 | 89.14 | 77.50 | 78.82 | 83.11 | 78.21 | 80.56 | 86.73 |
| Semantic | 89.13 | 90.82 | 92.15 | 83.88 | 84.01 | 87.93 | 83.15 | 78.84 | 86.99 |
| Atomic | 87.67 | 88.84 | 91.54 | 83.46 | 86.24 | 88.84 | 79.00 | 79.80 | 84.79 |
| **ToolWeaver** | **91.16** | **91.14** | **93.48** | **89.76** | **89.70** | **91.80** | **88.00** | **85.80** | **90.12** |

Table 7: End-to-end task completion performance (SoPR/SoWR) for different tokenization methods. All methods shown generate tool tokens directly, without the need for a retriever. The superior retrieval accuracy of ToolWeaver translates directly into higher task success rates.

| Tokenization | SoPR | | | | | | SoWR | | | | | |
|---|---|---|---|---|---|---|---|---|---|---|---|---|
| | I1 | I2 | I3 | I1-Tool. | I1-Cat. | I2-Cat. | I1 | I2 | I3 | I1-Tool. | I1-Cat. | I2-Cat. |
| Numerical | 21.98 | 9.12 | 11.20 | 20.68 | 26.14 | 17.20 | 16.56 | 16.04 | 16.39 | 20.89 | 23.53 | 14.52 |
| Hierarchical | 39.16 | 20.28 | 17.49 | 36.29 | 31.81 | 14.92 | 29.45 | 28.30 | 26.23 | 29.11 | 24.83 | 14.52 |
| Semantic | 50.20 | 28.91 | 16.39 | 33.02 | 51.42 | 27.02 | 39.26 | 29.24 | 32.79 | 29.11 | 43.79 | 22.58 |
| Atomic | 52.97 | **45.13** | 36.34 | 45.36 | 55.56 | 45.56 | 36.20 | 42.45 | 49.18 | 32.91 | 42.48 | **37.90** |
| **ToolWeaver** | **53.17** | 44.03 | **52.19** | **54.85** | **57.41** | **46.24** | **40.49** | **48.11** | **59.02** | 36.08 | 43.14 | 35.48 |

**Atomic Tokenization.** This is a widely-used baseline in generative tool-use models (Wang et al., 2024b). Each tool is represented by a single, unique special token. Specifically, the API function "compress" from the RESTful API "IMAGON" is tokenized into a single composite token like `<<IMAGON&&compress>>`. These new tokens are added to the LLM's vocabulary. While simple, this approach suffers from a linear growth in vocabulary size and fails to capture any semantic or collaborative relationships between tools, as their representations are learned independently. The results for this baseline are adopted from our ToolGen implementation.

**Numerical Tokenization.** This serves as a simple, non-semantic baseline. Each tool is mapped to a unique numeric string of fixed length. For a library of 47,000 tools, a five-digit string is used. For example, the 3rd tool in the corpus is represented as 0 0 0 0 3. This method creates a very small vocabulary overhead (only 10 digit tokens) but provides no semantic or structural priors to the model, forcing it to learn tool meanings from scratch.

**Hierarchical Tokenization.** This method adopts the hierarchical coding scheme from prior work (Wang et al., 2024b). Each tool is represented by a path in a pre-defined hierarchical structure,

resulting in a sequence of numerical codes (e.g., 1 0 1 4 0). This approach provides a structural prior by grouping related tools. However, since the hierarchy is based on static features, it may not fully capture the dynamic, collaborative relationships required for complex downstream tasks.

**Semantic Tokenization.** This approach uses human-readable, semantically meaningful parts of the tool's name or function as its representation. Instead of creating abstract IDs, it directly tokenizes the API's function name. For instance, an API function named `compress_for_imagon` would be represented by the sequence of its natural language tokens. This method leverages the LLM's existing linguistic knowledge but may struggle with APIs that have non-descriptive or ambiguous names. It also does not explicitly model the relationships between different tools.

### B.3 FULL RESULTS ON QWEN MODELS

To demonstrate the generalizability and robustness of the ToolWeaver framework beyond a single model architecture, we conducted supplementary experiments using the Qwen-2.5 model family, with the 1.5B, 3B, 7B and 14B parameter versions. We replicated our tool retrieval evaluation, comparing ToolWeaver directly against the strong generative baseline, ToolGen, which employs the "one-token-per-tool" paradigm.

The comprehensive results are presented in Table 8. The findings consistently show that ToolWeaver outperforms ToolGen across all tested model sizes and evaluation settings. Notably, the performance advantage of ToolWeaver is most pronounced in the more complex, multi-tool scenarios (I2 and I3), reinforcing our core claim that the collaborative-aware tokenization is crucial for sophisticated reasoning. This trend holds across different model scales. While the performance gap narrows slightly as model size increases, ToolWeaver maintains a consistent edge, highlighting the fundamental efficiency of its structured, collaborative-aware tokenization.

Table 8: Tool retrieval evaluation performance comparison between ToolGen and ToolWeaver across different Qwen-2.5 model sizes. Performance is measured by NDCG@k across different query complexities (I1, I2, I3).

| Method | I1 | | | I2 | | | I3 | | |
|---|---|---|---|---|---|---|---|---|---|
| | NDCG@1 | NDCG@3 | NDCG@5 | NDCG@1 | NDCG@3 | NDCG@5 | NDCG@1 | NDCG@3 | NDCG@5 |
| **Qwen-2.5-1.5B** | | | | | | | | | |
| ToolGen | 88.00 | 89.55 | **91.98** | 84.96 | 84.40 | 88.15 | 69.00 | 71.15 | 80.82 |
| **ToolWeaver** | **89.67** | **89.99** | 91.71 | **88.22** | **87.73** | **89.34** | **87.00** | **87.58** | **89.90** |
| **Qwen-2.5-3B** | | | | | | | | | |
| ToolGen | 90.33 | 90.32 | 93.04 | 85.21 | 84.29 | 88.58 | 85.00 | 81.10 | 87.76 |
| **ToolWeaver** | **90.67** | **91.66** | **92.99** | **88.47** | **89.08** | **90.28** | **88.00** | **87.63** | **90.95** |
| **Qwen-2.5-7B** | | | | | | | | | |
| ToolGen | 91.83 | 92.31 | **94.38** | 89.22 | 88.22 | 91.74 | 80.00 | 82.98 | 86.58 |
| **ToolWeaver** | **92.50** | **92.89** | 93.98 | **91.23** | **90.49** | **91.89** | **85.00** | **88.52** | **90.73** |
| **Qwen-2.5-14B** | | | | | | | | | |
| ToolGen | 90.66 | 91.55 | **93.84** | 89.22 | 88.56 | 91.36 | 82.00 | 81.79 | 88.28 |
| **ToolWeaver** | **91.00** | **91.97** | 93.22 | **91.23** | **90.34** | **91.97** | **85.00** | **83.16** | **88.46** |

### B.4 EXTENDED ANALYSIS ON GENERAL LANGUAGE CAPABILITIES

To investigate the impact of large-scale vocabulary expansion on an LLM's foundational abilities, we evaluated model performance on a suite of general NLP benchmarks. Integrating a vast toolset of nearly 47,000 APIs presents a critical trade-off between task-specific specialization and the preservation of an LLM's general language capabilities. We assessed this impact across three dimensions: general understanding, language modeling distribution, and text generation quality.

### B.4.1 EVALUATION SETUP

All evaluations were conducted using the open-source Language Model Evaluation Harness (Gao et al., 2024a), version 0.4.3, ensuring standardized prompting and scoring. For **Language Modeling**, we employed a sliding window approach with a window size and stride of 2,048 tokens

(non-overlapping) using the base model's tokenizer. For **Text Summarization**, to ensure efficiency, we evaluated a random subset of 500 samples for each task. We report ROUGE scores for n-gram overlap and BERTScore (F1) using "roberta-large" for semantic similarity.

### B.4.2 Benchmark Descriptions

We used a diverse set of benchmarks to evaluate the models:

- **MMLU** (Massive Multitask Language Understanding) (Hendrycks et al., 2021): A comprehensive benchmark covering 57 subjects to test world knowledge and problem-solving ability.
- **BoolQ** (Boolean Questions) (Clark et al., 2019): A reading comprehension dataset consisting of yes/no questions.
- **PIQA** (Physical Interaction Question Answering) (Bisk et al., 2020): A commonsense reasoning benchmark testing understanding of everyday physical situations.
- **HellaSwag** (Zellers et al., 2019): A commonsense inference task that challenges models to choose the most plausible completion for a given text context.
- **OpenBookQA** (Mihaylov et al., 2018): A science question-answering dataset requiring reasoning with a small set of common knowledge facts. For this benchmark, we report normalized accuracy.
- **WinoGrande** (Sakaguchi et al., 2020): A commonsense reasoning dataset focused on pronoun resolution, designed to be robust against statistical biases.
- **WikiText-2** (Merity et al., 2016): A standard language modeling benchmark. We use the validation split to measure Perplexity (PPL) and Negative Log-Likelihood (NLL).
- **CNN/DailyMail** (See et al., 2017): An abstractive summarization dataset consisting of news articles.
- **XSum** (Narayan et al., 2018): A dataset requiring highly abstractive, single-sentence summaries from BBC articles.

### B.4.3 Full Experimental Results

**General Understanding Benchmarks.** The results for tasks such as MMLU, BoolQ, and PIQA are presented in Table 9. The data shows that the "one-token-per-tool" approach, embodied by Tool-Gen, comes at a catastrophic cost to the model's core competencies. Its average performance plummets by nearly 23 points (from 66.81% to 43.87%) compared to the original Llama-3-8B model. In stark contrast, ToolWeaver demonstrates far more effective management of this trade-off. While specialization still incurs a cost, our logarithmically scaled vocabulary results in a much more contained degradation of only 8.4 points. Crucially, this means ToolWeaver mitigates over 63% of the performance loss seen with the naive generative approach.

Table 9: Performance on general NLP benchmarks. Scores are accuracy (%). For OpenBookQA, the score represents normalized accuracy.

| Model | MMLU | BoolQ | PIQA | HellaSwag | OpenBookQA* | WinoGrande | Avg. |
|---|---|---|---|---|---|---|---|
| Llama-3-8B (Base) | 62.19 | 81.10 | 79.43 | 60.07 | 45.00 | 73.09 | 66.81 |
| ToolGen | 23.52 | 62.17 | 60.07 | 31.60 | 31.00 | 54.85 | 43.87 |
| ToolWeaver | 41.93 | 78.20 | 74.54 | 51.19 | 38.40 | 65.98 | 58.37 |

**Language Modeling Distribution.** To evaluate the integrity of the model's probability distribution, we report the perplexity on WikiText-2 in Table 10. ToolGen exhibits a severe explosion in perplexity, reaching 104.54, which indicates a significant disruption to the natural language distribution likely caused by the massive injection of initialized tokens. Conversely, ToolWeaver maintains a much lower perplexity of 25.36. This result suggests that our structured, collaborative-aware codes integrate more harmoniously with the pre-trained weights, preserving the model's ability to predict natural language sequences.

Table 10: Language modeling evaluation on WikiText-2 (Validation Split).

| Model | Avg NLL | Perplexity |
|---|---|---|
| Llama-3-8B (Base) | 1.847 | 6.34 |
| **ToolWeaver** | 3.233 | 25.36 |
| ToolGen | 4.650 | 104.54 |

**Text Generation Quality.** We assessed generation capabilities via zero-shot summarization, as detailed in Table 11. On the CNN/DailyMail dataset, ToolWeaver performs nearly on par with the Base LLM (BERTScore 0.8507 vs. 0.8535). On the more challenging XSum dataset, which requires high-level abstraction, ToolGen suffers a notable drop in precision (ROUGE-2 drops to 0.0175). In comparison, ToolWeaver retains robust generation capabilities (ROUGE-2 0.0261). These results confirm that ToolWeaver not only preserves understanding but also maintains the ability to generate coherent and accurate text, a capability that is often compromised in standard generative tool learning methods.

Table 11: Zero-shot summarization performance (Means over 500 samples).

| Model | CNN/DailyMail | | | | XSum | | | |
|---|---|---|---|---|---|---|---|---|
| | BERTScore | R-1 | R-2 | R-L | BERTScore | R-1 | R-2 | R-L |
| Llama-3-8B (Base) | 0.8535 | 0.2107 | 0.0856 | 0.1501 | 0.8505 | 0.1494 | 0.0376 | 0.1060 |
| ToolGen | 0.8293 | 0.1541 | 0.0535 | 0.1127 | 0.8253 | 0.0969 | 0.0175 | 0.0702 |
| **ToolWeaver** | **0.8507** | **0.2021** | **0.0813** | **0.1461** | **0.8418** | **0.1240** | **0.0261** | **0.0872** |

## B.5 Codebook Hyperparameter Sensitivity Analysis

To empirically validate the design choices of ToolWeaver, we conduct a sensitivity analysis on two critical hyperparameters governing the structured tokenization: the total size of the added vocabulary and the depth of the hierarchical code (code length).

**Impact of Vocabulary Size.** We first investigate how the total number of added tokens affects retrieval performance. In this experiment, we maintain a fixed code length of $L = 2$ while varying the size $K$ of the codebooks at each layer. We specifically evaluated configurations with equal layer sizes of $K \in \{512, 1024, 2048, 5096\}$, which correspond to total added vocabulary sizes of 1,024, 2,048, 4,096, and 10,192 tokens, respectively. As illustrated in Figure 5(a), the Average NDCG exhibits an inverted U-shape, peaking at the $1024 \times 2$ configuration (2,048 tokens). When the codebook is too small ($512 \times 2$), performance is suboptimal, likely due to code collision where functionally distinct tools are forced to map to the same identifiers. Crucially, as we increase the size to $2048 \times 2$ and further to $5096 \times 2$ (totaling 10,192 tokens), we observe a significant performance drop. This decline confirms our hypothesis regarding the sparsity of collaborative signals: as the vocabulary grows towards the "one-token-per-tool" extreme, the probability of related tools sharing a common code decreases. This dilutes the dense co-occurrence patterns ToolWeaver relies on. Consequently, our default setting of $1024 \times 2$ strikes the optimal balance between representational capacity and signal density.

**Impact of Code Length.** Next, we evaluate the effect of the code sequence length $L$, which corresponds to the depth of the quantization hierarchy. We fix the codebook size per layer at $K = 1024$ and vary $L$ from 2 to 6. Figure 5(b) shows that performance improves significantly as the hierarchy deepens, reaching a peak at $L = 4$. This trend suggests that a deeper hierarchy captures finer-grained semantic nuances, aiding in precise tool disambiguation. However, performance begins to degrade as the length extends beyond 4 layers, dropping to 90.82 at $L = 6$. We attribute this degradation primarily to the increased difficulty of autoregressive generation, where longer sequences heighten the risk of error propagation during decoding, and potentially to the diminishing returns of residual quantization at deeper layers. Although $L = 4$ offers the highest theoretical performance, we utilize $L = 2$ in our main experiments to maintain a favorable trade-off between retrieval accuracy and inference efficiency.

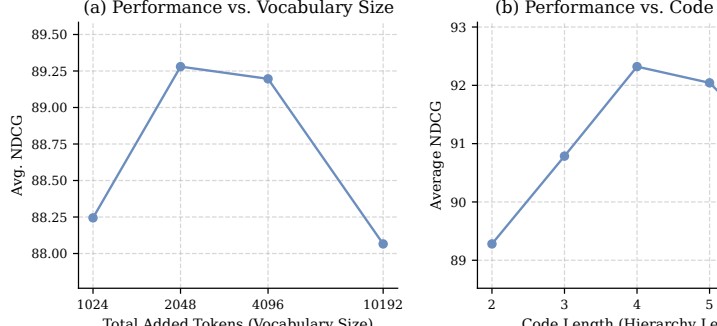

Figure 5: Hyperparameter sensitivity analysis. **(a) Performance vs. Vocabulary Size:** Evaluated with fixed code length $L = 2$. Performance peaks at 2,048 tokens, confirming that a compact vocabulary fosters better collaborative learning than a sparse, large one. **(b) Performance vs. Code Length:** Evaluated with fixed codebook size $K = 1024$. While deeper hierarchies ($L = 4$) improve semantic resolution, excessively long sequences ($L = 6$) degrade performance due to generation complexity.

### B.6 SINKHORN-KNOPP EFFICIENCY AND STABILITY ANALYSIS

To ensure that the conflict mitigation mechanism via uniform mapping does not introduce computational bottlenecks or numerical instability, we conducted a comprehensive profiling analysis of the Sinkhorn-Knopp algorithm during the training of ToolWeaver.

#### B.6.1 EXPERIMENTAL SETUP

We profiled the training process on a single NVIDIA A100-SXM4-80GB GPU, using the full Tool-Bench embedding matrix ($46,984 \times 768$) and a batch size of $B = 5,096$, consistent with our main experiments described in Appendix A.3. The model employs two codebooks ($L = 2$) with 1,024 codes each. In this configuration, the Sinkhorn-Knopp algorithm is invoked exactly twice per optimizer step (once for each codebook's assignment) to enforce uniformity constraints. To analyze the trade-off between stability and cost, we fixed the number of Sinkhorn iterations at 50 and swept the entropy regularization parameter $\epsilon$ across values of $\{0.005, 0.01, 0.02\}$.

#### B.6.2 RESULTS AND DISCUSSION

**Computational Cost Analysis.** We analyzed the steady-state training time per step after the initial warm-up phase. The average total time per training step is $0.161$ seconds. Of this, the two Sinkhorn solves consume a combined average of $0.0283 \pm 0.0002$ seconds (approximately $14.15$ ms per call). This corresponds to a computational overhead of only **17.6%** of the total step time. The vast majority of the computation ($\sim 82\%$) is dedicated to the encoder/decoder MLP layers and the backward pass. These results empirically verify that the Sinkhorn integration is computationally efficient and does not constitute a bottleneck for training scalability.

**Stability and Uniformity.** Numerical stability is critical for optimal transport algorithms. Throughout our profiling of 20 consecutive batches, we observed no NaN or Inf values, confirming the numerical robustness of our implementation. Regarding uniformity, with our chosen setting of $\epsilon = 0.01$, the final residual assignments remain highly balanced. The standard deviation of tool counts per code is $1.38$ (against a theoretical target of $4.97$ tools per code), with $95\%$ of codes receiving between 3 and 7 assignments per batch. This indicates that the algorithm successfully mitigates index collapse without enforcing an overly rigid permutation.

**Ablation on Entropy Regularization ($\epsilon$).** We further analyzed the impact of $\epsilon$ on performance. As summarized in Table 12, $\epsilon = 0.01$ provides the optimal balance. Strict regularization ($\epsilon = 0.005$) sharpens the distribution (Rel. Std 0.19) but increases overhead to 20.3% due to slower convergence. Conversely, loose regularization ($\epsilon = 0.02$) degrades uniformity (Rel. Std 0.47) without improving runtime. This justifies our choice of $\epsilon = 0.01$ for the main experiments.

Table 12: Profiling results for Sinkhorn-Knopp at varying entropy regularization levels ($\epsilon$). The chosen setting ($\epsilon = 0.01$) provides the best trade-off between runtime overhead and uniformity.

| $\epsilon$ Setting | Step Overhead (%) | Time per Call (ms) | Uniformity (Rel. Std) | Conclusion |
|---|---|---|---|---|
| 0.005 (Strict) | 20.3% | 18.9 | **0.19** | Slower convergence |
| **0.01 (Ours)** | **17.6%** | **14.2** | 0.28 | **Optimal balance** |
| 0.02 (Loose) | 17.5% | 14.1 | 0.47 | Degraded uniformity |

```
{
  "product_id": "api_2c3bbf59-df39-4b01-b91b-0f176c8effd9",
  "tool_description": "Extract the information on a Thai driver's license and return text results such as driver's license number and personal information.",
  "home_url": "https://rapidapi.com/the-brainstem-brainbotapi/api/thai-drivers-license-ocr/",
  "name": "Thai Drivers License OCR",
  "title": "Thai Drivers License OCR",
  "pricing": "FREEMIUM",
  "tool_name": "Thai Drivers License OCR",
  "score": null,
  "host": "thai-drivers-license-ocr.p.rapidapi.com",
  "api_list": [
    {
      "name": "Driver's  License",
      "url": "https://thai-drivers-license-ocr.p.rapidapi.com/api/v1/ocr-licensedriver",
      "description": "Extract the information on a Thai driver's license and return text results such as driver's license number and personal information.",
      "method": "POST",
      "required_parameters": [],
      "optional_parameters": [],
      "code": "import requests\n\nurl = \"https://thai-drivers-license-ocr.p.rapidapi.com/api/v1/ocr-licensedriver\"\n\nheaders = {\n        \"X-RapidAPI-Key\": \"SIGN-UP-FOR-KEY\",\n        \"X-RapidAPI-Host\": \"thai-drivers-license-ocr.p.rapidapi.com\"\n    }\n\nresponse = requests.post(url, headers=headers)\nprint(response.json())\n",
      "convert_code": "import requests\n\nurl = \"https://thai-drivers-license-ocr.p.rapidapi.com/api/v1/ocr-licensedriver\"\n\nheaders = {\n        \"X-RapidAPI-Key\": \"SIGN-UP-FOR-KEY\",\n        \"X-RapidAPI-Host\": \"thai-drivers-license-ocr.p.rapidapi.com\"\n    }\n\nresponse = requests.post(url, headers=headers)\nprint(response.json())\n",
      "test_endpoint": "",
      "statuscode": 200,
      "schema": {}
    }
  ],
  "category_name": "Video_Images"
}
```

Figure 6: A real RESTful API example for extracting information from a Thai driver's license, including details like the API's endpoint, parameters, and code snippet for implementation.

```
{
  "tool_name":"URL to QRCode Image API",
  "tool_description":"This API takes URL and return as a QR Code image",
  "title":"URL to QRCode Image API",
  "pricing":"FREEMIUM",
  "score":null,
  "home_url":"https://rapidapi.com/ohidur/api/url-to-qrcode-image-api/",
  "host":"url-to-qrcode-image-api.p.rapidapi.com",
  "api_list":[
    {
      "name":"QR Code image",
      "url":"https://url-to-qrcode-image-api.p.rapidapi.com/qr",
      "description":"This endpoint takes a 'GET' request with url or string as a parameter and returns QR code image",
      "method":"GET",
      "required_parameters":[],
      "optional_parameters":[
        {
          "name":"url",
          "type":"STRING",
          "description":"",
          "default":"https://www.google.com"
        }
      ],
      "code":"import requests\n\nurl = \"https://url-to-qrcode-image-api.p.rapidapi.com/qr\"\n\nquerystring = {\"url\": url}\n\nheaders = {\n  \"X-RapidAPI-Key\": \"SIGN-UP-FOR-KEY\",\n  \"X-RapidAPI-Host\": \"url-to-qrcode-image-api.p.rapidapi.com\"}\n\nresponse = requests.get(url, headers=headers, params=querystring)\n\nprint(response.json())\n",
      "statuscode":111,
      "body":"",
      "headers":"",
      "schema":""
    }
  ]
}
```

Figure 7: A real tool example. It shows the details of the "URL to QRCode Image API", including its description, endpoint, method, parameters, and a code snippet for implementation.

## B.7 INFERENCE LATENCY AND MEMORY ANALYSIS

While ToolWeaver achieves logarithmic scalability regarding vocabulary size, representing a single tool as a sequence of $L$ codes naturally introduces more decoding steps compared to the atomic "one-token-per-tool" approach used in baselines like ToolGen. To rigorous assess the practical cost of this design, we conducted a systematic evaluation of decoding latency, throughput, and memory consumption.

### B.7.1 EXPERIMENTAL SETUP

We measured the inference performance on a single NVIDIA A100-80GB GPU. To ensure a fair comparison, we evaluated both ToolGen (Atomic) and ToolWeaver with varying codebook depths ($L \in \{2, 3, 4\}$) and a fixed codebook size of $K = 1024$. The metrics were averaged across the I1, I2, and I3 retrieval tasks to cover varying query complexities. We report:

- **Avg Latency (ms):** The average wall-clock time required to decode the tool identifier(s) for a single query.
- **P95 Latency (ms):** The 95th percentile latency, reflecting worst-case performance.
- **Avg Throughput (Tok/s):** The number of tokens generated per second during the decoding phase.
- **Peak GPU Memory (GB):** The maximum GPU memory allocated during inference.

### B.7.2 RESULTS AND DISCUSSION

The results are summarized in Table 13. We observe the following trends:

Table 13: Inference efficiency comparison. We compare the atomic baseline (ToolGen) against ToolWeaver with increasing codebook depths ($L$). While latency increases linearly with depth due to longer sequence generation, the absolute overhead is minimal ($\sim$ 20-75ms). Crucially, ToolWeaver maintains a lower and constant memory footprint.

| Model Configuration | Representation Structure | Avg Latency (ms) | P95 Latency (ms) | Throughput (Tok/s) | Peak Memory (GB) |
|---|---|---|---|---|---|
| ToolGen (Baseline) | Atomic (1-level) | 108.16 | 111.98 | 19.54 | 15.77 |
| **ToolWeaver** ($L = 2$) | [1024, 1024] | 128.21 | 132.43 | 24.53 | **15.08** |
| **ToolWeaver** ($L = 3$) | [1024, 1024, 1024] | 157.65 | 165.73 | 26.34 | 15.10 |
| **ToolWeaver** ($L = 4$) | [1024, 1024, 1024, 1024] | 183.14 | 189.11 | 28.26 | 15.11 |

**Latency Trade-off is Acceptable.** As expected, latency increases linearly with the number of codebook layers. Comparing the standard $L = 2$ setting of ToolWeaver to ToolGen, the overhead per query is approximately 20ms (108.16ms vs. 128.21ms). Even with a deeper hierarchy ($L = 4$), the total latency remains under 200ms. In the context of tool-augmented agents, where executing an external API call typically consumes hundreds of milliseconds to seconds, this decoding overhead is negligible. This confirms that the trie-constrained decoding over a hierarchical code space is highly efficient for online deployment.

**Higher Token Throughput.** Interestingly, ToolWeaver exhibits higher token throughput (Tok/s) as $L$ increases. This is a natural consequence of the hierarchical representation: decoding a single logical tool requires generating $L$ simpler code tokens. Since the computational cost per step is dominated by the transformer's forward pass (which remains constant), generating a sequence of cached code-tokens allows the system to amortize the overhead, resulting in higher apparent throughput (19.54 vs. 28.26 Tok/s). This indicates that the model's generation speed is not bottlenecked by the codebook lookup.

**Memory Efficiency.** A significant advantage of ToolWeaver is its memory efficiency. ToolGen requires maintaining a massive embedding table and LM head for nearly 47,000 atomic tool tokens, resulting in a peak memory usage of ~15.77 GB. In contrast, ToolWeaver reduces the vocabulary

expansion to a logarithmic scale (e.g., $2 \times 1024$ tokens), keeping the peak memory stable at $\sim$15.10 GB across all settings. This saving of approximately 0.67 GB is substantial for deploying LLMs on memory-constrained edge devices, validating our claim that ToolWeaver is a more scalable solution for massive tool libraries.

### B.8 FAILURE ANALYSIS

To investigate the limitations of ToolWeaver, we analyzed error cases in the end-to-end evaluation. We categorize the **first occurring error** in a trajectory using a strict hierarchical logic: First, we check Process Consistency; if the predicted step index exceeds or falls short of the ground truth length, it is labeled as **Redundant** or **Incomplete Process**, respectively. Second, if the length is valid but the predicted identifier mismatches the ground truth, it is marked as **Wrong Tool**. Finally, if the tool is correct but fails due to parsing errors, missing fields, or runtime exceptions, it is categorized as **Wrong Parameters**. Figure 8 illustrates the distribution of these errors.

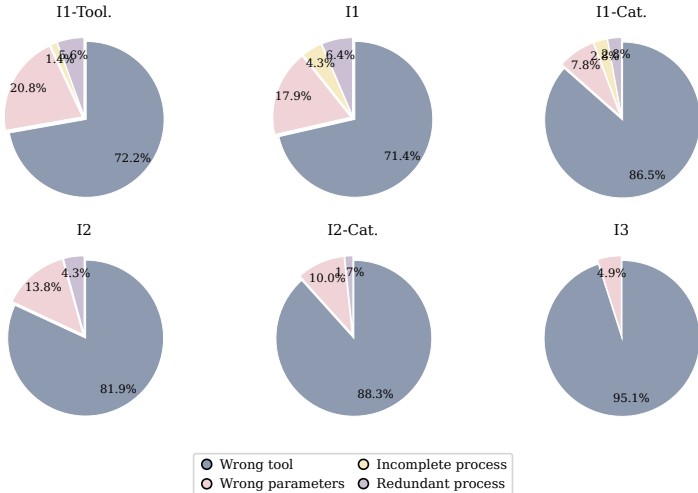

Figure 8: Distribution of failure types across ToolBench scenarios (I1-I3) and generalization splits (Tool/Cat).

The distribution reveals a clear shift in failure modes across different stages of complexity. Wrong Tool (Blue) is the dominant error, increasing significantly from I1 ( 71%) to I3 ( 95%). This indicates that as the reasoning chain grows longer and more complex (I3), the primary bottleneck becomes the precise retrieval of the correct API from the massive 47k corpus, rather than planning length. Consequently, Wrong Parameters (Pink) shows a stable presence (10–20%) in simpler scenarios (I1/I2), suggesting that once the model successfully locates the correct tool, its ability to comprehend API schemas and generate valid arguments is relatively robust. However, in I3, the parameter error rate drops artificially (4.9%) simply because the model rarely passes the initial tool selection check. Similarly, Incomplete/Redundant processes are visible in simpler tasks but vanish in complex ones, confirming that in multi-step scenarios, the agent struggles primarily with semantic discrimination of tools before it can even exhibit planning or formatting faults.

## C QUALITATIVE ANALYSIS AND EXAMPLES

### C.1 CASE STUDY: ANALYSIS OF LEARNED TOOL CODES AND MULTI-TOOL COORDINATION

This section provides a detailed qualitative analysis of the hierarchical codes learned by ToolWeaver. By examining the tool clusters formed under specific high-level codes, we interpret their emergent semantic meaning. Furthermore, we analyze a real execution trajectory to illustrate how these shared parent codes facilitate complex multi-tool coordination.

### C.1.1 Detailed Analysis of Learned Structures

We observe that the model learns meaningful abstractions ranging from service encapsulation to functional decomposition. Below we detail three specific cases:

**Case 1: Clear Service Encapsulation (`<T1_747>`).** This high-level code has learned to cleanly encapsulate tools related to the video game "Guild Wars 2". An analysis of the 97 tools sharing this primary code confirms this, showing a perfect 100% alignment with the ground-truth *Gaming* category. Representative tools include:

- `Get Account Info` (from the "Guild Wars 2" service)
- `Get Achievements` (from the "Guild Wars 2" service)
- `Get Character Hero Points` (from the "Guild Wars 2" service)
- `Get Pvp Stats` (from the "Guild Wars 2" service)

While this grouping alone demonstrates strong semantic clustering, its true value lies in creating the prerequisite for learning collaboration. By sharing the parent code `<T1_747>`, these tools provide a dense, shared signal during training. This allows the model to efficiently learn that these tools are often co-utilized to answer complex queries about the game, overcoming the signal sparsity issue inherent in methods that use monolithic, independent tool IDs.

**Case 2: Hierarchical Functional Decomposition (`<T1_184>`).** This case highlights the model's ability to perform functional decomposition, grouping tools related to Billboard music charts. All 92 tools under this code correctly belong to the *Music* category. The model correctly groups tools like `Hot 100` (from the "Billboard" service) and `New Zealand Songs` (from the "Billboard API" service) under a single high-level code, demonstrating that it learns a true functional hierarchy that transcends superficial metadata. Representative tools include:

- `Hot 100` (from the "Billboard" service)
- `New Zealand Songs` (from the "Billboard API" service)
- `Billboard Japan Hot 100` (from the "Billboard API" service)
- `Artist 100` (from the "Billboard" service)

This learned structure provides a robust foundation for complex reasoning. The shared parent code `<T1_184>` acts as a strong collaborative prior, signaling to the model that these distinct chart APIs can be orchestrated to fulfill a multifaceted request (e.g., comparing charts across regions). This is a clear example of how our collaborative-aware tokenization creates a meaningful structure that is essential for enabling complex, multi-step planning.

**Case 3: Coherent Semantic Grouping for Coordination (`<T1_996>`).** We further observe that `<T1_996>` successfully aggregates distinct tools related to *Health & Fitness Metrics*. This parent code clusters various metabolic calculation tools by function. Representative tools include:

- `BMI Calculator v2` (for standard BMI calculation)
- `BMI v2` (handling specific inputs like metric vs. imperial units)
- `BMR & TMR` (for metabolic rate calculations)

By organizing these functionally similar but operationally distinct tools under a coherent "Health Metrics" family, the model establishes a stable semantic anchor in the latent space. This shared parent code serves as a critical navigational aid, allowing the model to effectively pivot between related tool variations when specific input requirements (e.g., units of measurement) change during a task.

### C.1.2 Multi-Tool Coordination in Trajectories

To explicitly demonstrate how the coherent grouping described in **Case 3** facilitates actual task execution, Figure 9 presents a real execution trajectory.

In this scenario, the user requests a comprehensive integration guide for a BMI app, requiring the use of multiple distinct BMI calculators. The trajectory reveals that the model consistently utilizes tools within the `<T1_996>` family defined above. It first invokes the standard BMI tool (`<T1_996><T2_258>`). Realizing the need for versatile inputs, it seamlessly pivots to a metric-specific endpoint (`<T1_996><T2_606>`) and then to an imperial-unit endpoint (`<T1_996><T2_328>`). This behavior demonstrates that the shared parent code acts as a collaborative bridge, allowing the model to effectively "explore" related sub-functions (the child codes) without losing the high-level context of the task.

| **Multi-Tool Coordination Trajectory (BMI Integration Task)** | |
|---|---|
| **User Query** | I'm designing a mobile app for tracking body mass index, and I need to integrate the BMI calculation API. Could you provide me with the necessary API endpoints and parameters? |

*Step 1: Retrieve Standard Calculator*

| | |
|---|---|
| **Thought** | The user needs BMI endpoints. I will start by retrieving the standard calculator to cover the basic use case. |
| **Action** | **Tool:** `bmi_for_bmi_calculator_v2` |
| | **Code: `<T1_996>`**`<T2_258>`  *(Family: Health Metrics)* |
| | **Args:** `{"weight": 70, "height": 170}` |

*Step 2: Explore Metric-Specific Variation*

| | |
|---|---|
| **Thought** | To provide a comprehensive guide, I should also check for endpoints that specifically accept explicit metric units to ensure precision. |
| **Action** | **Tool:** `gives_the_bmi_when_you_input_values_in_metric_units` |
| | **Code: `<T1_996>`**`<T2_606>`  *(Same Parent Code)* |
| | **Args:** `{"heightincm": 170, "weightinkg": 70}` |

*Step 3: Explore Imperial-Unit Variation*

| | |
|---|---|
| **Thought** | Finally, I need to cover users who prefer imperial units (feet and inches) to make the app integration guide complete. |
| **Action** | **Tool:** `height_in_feet_and_inches_input_weight_in_kilograms` |
| | **Code: `<T1_996>`**`<T2_328>`  *(Same Parent Code)* |
| | **Args:** `{"weightinkg": 70, "heightfeet": 5, "heightinches": 10}` |
| **Answer** | Here are the necessary API endpoints for your BMI app: 1. Standard: `bmi_...` 2. Metric: `gives_the_bmi...` 3. Imperial: `height_in_feet...` |

Figure 9: A real execution trajectory demonstrating intra-family coordination. The model leverages the shared parent code **`<T1_996>`** (Health Metrics) to coherently navigate between three distinct but functionally related BMI tools (Standard, Metric, Imperial) to fulfill a complex user request.

## C.2 EXAMPLES FOR TOOLS AND APIS

To illustrate the diversity and realism of the tools and APIs used in our experiments, we present two representative examples from the ToolBench dataset. As shown in Figure 6, the first example demonstrates a real-world RESTful API for extracting information from a Thai driver's license, including its endpoint, required parameters, and a sample implementation snippet. Figure 7 showcases a functional tool that converts URLs into QR code images, highlighting its usage description, supported parameters, and implementation code. These examples reflect the practical and executable nature of the APIs included in ToolBench.

## C.3 EXAMPLES OF ALIGNMENT TUNING DATA

We also present examples for our two-stage generative alignment process. Figure 10 shows a training instance for tool retrieval alignment, where the model learns to generate a tool's symbolic identifier

| Retrieval Training |
|---|
| **User:**
My friend is a big fan of TV shows and wants to know the most popular TV shows right now. Can you fetch the list of most popular TV shows? Also, provide the overview details of these shows, such as their running time and number of episodes.
**Assistant:**
`{Tool Identifier for "Title Get Overview Details"}` |

Figure 10: Dataset example for tool selection in retrieval training. We use user role to represent queries and assistant role to represent the chosen tool invocation. The placeholder {`Tool Identifier ...`} represents the target output, whose format varies depending on the specific tokenization method being trained.

from a query. Figure 11 demonstrates a training instance for tool usage trajectory alignment. It shows a multi-step interaction where the agent reasons, invokes tools, and recovers from failures to complete the task.

### C.4 REAL END-TO-END INFERENCE TRAJECTORY

To further illustrate how ToolWeaver behaves at inference time, Figure 12 presents a real end-to-end tool-use trajectory on a complex, multi-part ToolBench-style query. The user asks the agent to design a treasure hunt game that uses QR codes embedding riddles, per-clue identifiers, and a global UUID for the entire game session. For clarity, we omit the verbose system prompt and only show the user and assistant turns, as well as the intermediate action and argument tokens generated by the model.

## D  THE USAGE OF LLMS

LLMs were used as an assistive tool to improve the grammar and clarity of the manuscript, help summarize related literature, and refine concepts for figures. The core research ideas, experimental design, and conclusions were developed entirely by the human authors, who take full responsibility for the final content of this paper.

| End-to-End Agent-Tuning |
|---|
| **System:** You are AutoGPT, you can use many tools (functions) to do the following task. 
 First I will give you the task description, and your task start. 
 At each step, you need to give your thought to analyze the status now and what to do next, with a function call to actually execute your step. 
 After the call, you will get the call result, and you are now in a new state. 
 Then you will analyze your status now, then decide what to do next... 
 After many (Thought-call) pairs, you finally perform the task, then you can give your final answer. 
 **Remember:** 
 1. The state change is irreversible, you can't go back to one of the former states. If you want to restart the task, say "I give up and restart". 
 2. All the thought is short, at most in 5 sentences. 
 3. You can do more than one try, so if your plan is to continuously try some conditions, you can do one of the conditions per try. 
 **Let's Begin!** 
 **Task description:** You should use functions to help handle the real time user queries. Remember: 
 1. ALWAYS call `Finish` function at the end of the task. And the final answer should contain enough information to show to the user. 
   If you can't handle the task, or you find that function calls always fail (the function is not valid now), use function `Finish->give_up_and_restart`. 
 2. Only generate actions and inputs. |
| **User:** Are you able to help me sanitize some text that has profanity? I'm looking for a way to automatically remove any bad words. Can you clean up this sentence - 'this is some damn good text that needs cleaning. Clearly its pretty crap.'? Begin! |
| **User:** This is not the first time you try this task, all previous trails failed. 
 Before you generate my thought for this state, I will first show you your previous actions for this state, and then you must generate actions that is different from all of them. 
 Here are some previous actions candidates: 
 `["name": "{Tool Identifier for Remove Profanity}", "arguments": "\n \\"text\\": \\"this is some damn good text that needs cleaning. Clearly its pretty crap.\\"", "function_output": "\\"error\\": \\"", "response": \\"'message': \\"API doesn't exists\\"\\""]` 
 Remember you are now in the intermediate state of a trail, you will first analyze the now state and previous action candidates, then make actions that is different from all the previous. |
| **Assistant:** Based on the previous action, it seems that the function call to `{Tool Identifier for Remove Profanity}` failed because the API does not exist. 
 To handle the profanity in the text, I will use a different approach. Instead of relying on an external API, I will manually create a list of profanity words and replace them with asterisks. 
 My new action will be to replace the profanity words in the text with asterisks. |
| **User:** Generate the action. |
| **Assistant:** `<<Finish>>` |
| **User:** Please give the input. Here is the documentation: 
 `{'required': [], 'optional': [{'name': 'give_answer', 'description': 'Output the answer', 'type': 'string', {'name': 'give_up_and_restart', 'description': 'Unable to handle the task from this step', 'type': 'string'}]}}` |
| **Assistant:** `{` 
 `  "return_type": "give_answer",` 
 `  "final_answer": "this is some **** good text that needs cleaning. Clearly its pretty ****."` 
 `}` |

Figure 11: An example for end-to-end agent-tuning. The placeholder `{Tool Identifier ...}` represents the target output, whose format varies depending on the specific tokenization method being trained.

| End-to-End Tool-Use Trajectory | |
|---|---|
| **System:** | Standard AutoGPT-style tool-using instructions (omitted here for brevity). |
| **User:** | I am planning a treasure hunt game for my family reunion and I want to use QR codes to provide clues. Generate QR codes with riddles as the data parameter for each clue. Additionally, include a random string of characters as a unique identifier for each QR code. Finally, generate a UUID for the treasure hunt game to ensure the uniqueness of the game session. |
| **Assistant:** | To generate QR codes with riddles as the data parameter for each clue, I first decide to call the "<T1_124><T2_781>" function, which corresponds to the `qr_code_gen_for_qr_api` tool. This tool converts a URL into a QR code image, and I will use it to quickly verify that I can generate a basic QR code for one of the clues. |
| **User:** | Generate the action. |
| **Assistant:** | `qr_code_gen_for_qr_api` |
| **User:** | Please give the input. Here is the documentation: `{'name': 'QR Code API (Web URL)', 'description': 'Call the API Endpoint /qr and pass in the url parameter...', 'required': [{'name': 'url', 'type': 'STRING'}], 'optional': [...]}` |
| **Assistant:** | `{"url": "https://example.com/clue1"}` |
| **Assistant:** | The basic URL-to-QR API works, but it only accepts a simple URL string and cannot directly encode richer structured data such as "riddle text + per-clue identifier". To embed both the riddle and a random string in the QR payload, I need a more flexible QR code generator that allows arbitrary `data` content and styling. For this, I will use the "<T1_93><T2_913>" function. |
| **User:** | Generate the action. |
| **Assistant:** | `create_qr_code_for_flowcode` |
| **User:** | Please give the input. Here is the documentation: `{'name': 'Generate Advance – Direct Image', 'description': 'Generates a QR code as a direct image with additional settings.', 'required': [{'name': 'data', 'type': 'STRING'}], 'optional': [...]}` |
| **Assistant:** | `{"data": "GameSession: FAMILY-REUNION-2025; ClueId: Z7QK9M; Riddle: What is the capital city of France?", "imageType": "png", "style": "default", "moduleShape": "circle", "rimTextTop": "Clue 1", "rimTextBottom": "Scan to reveal the riddle!"}` |
| **Assistant:** | To generate a UUID for the overall treasure hunt game session (distinct from the per-clue random string embedded in each QR code), I now need to call the "<T1_621><T2_2>" function. This corresponds to the `random_uuid_for_teamriverbubbles_random_utilities` tool, which returns a UUID v4 that I can use as a unique game identifier. |
| **User:** | Generate the action. |
| **Assistant:** | `random_uuid_for_teamriverbubbles_random_utilities` |
| **User:** | Please give the input. Here is the documentation: `{'name': 'Generate UUID', 'description': 'Generate UUID v4 with dash or not.', 'required': [], 'optional': [...]}` |
| **Assistant:** | `{}` |
| **Assistant:** | The UUID tool has returned a unique identifier for the treasure hunt game session, which I will associate with all generated QR codes. Each QR code embeds its own `ClueId` (e.g., `Z7QK9M`) inside the `data` field, while the session-level UUID serves as a global game identifier for tracking and management. |
| **User:** | Generate the action. |
| **Assistant:** | `<<Finish>>` |
| **User:** | Please give the input. Here is the documentation: `{'required': [], 'optional': [{'name': 'give_answer', 'description': 'Output the answer', 'type': 'string'}, {'name': 'give_up_and_restart', 'description': 'Unable to handle the task from this step', 'type': 'string'}]}` |
| **Assistant:** | `{` `"return_type": "give_answer",` `"final_answer": "I have generated a treasure-hunt QR code clue that embeds both a riddle and a per-clue identifier, and also created a UUID for the whole game session. Concretely, the advanced QR API is called with a data payload like 'GameSession: FAMILY-REUNION-2025; ClueId: Z7QK9M; Riddle: What is the capital city of France?', and rendered as a PNG image with circular modules, top rim text 'Clue 1', and bottom rim text 'Scan to reveal the riddle!'. Separately, the UUID tool returns a game-level identifier such as 'e286537a-2758-4144-8890-22a575b9157a', which uniquely tags this treasure hunt session."` `}` |

Figure 12: A real end-to-end inference trajectory of ToolWeaver handling a complex, multi-part user query. The system prompt is omitted for brevity. The agent successfully reasons, selects multiple tools via hierarchical codes, adapts its tool choice after an initial limitation, and synthesizes the final answer.

