# OpenReview forum: "ToolWeaver: Weaving Collaborative Semantics for Scalable Tool Use in Large Language Models"
_ICLR.cc/2026/Conference — ICLR 2026 Poster_

### Official Review · Reviewer_fdct · 2025-10-29

**Soundness:** 3
**Presentation:** 3
**Contribution:** 2
**Rating:** 4
**Confidence:** 4

**Summary:**

The paper proposes ToolWeaver, a novel generative tool retrieval framework that encodes tools into hierarchical sequences with collaborative signals between tools. The collaborative-aware tokenization process allows the framework to weave tools' intrinsic semantics with their extrinsic co-usage patterns, encouraging functionally related tools to share codes. Experiments on the ToolBench benchmark demonstrate that the framework significantly outperforms state-of-the-art methods on tool retrieval and end-to-end performance.

**Strengths:**

- Collaborative-aware tokenization process enables it to learn robust collaborative patterns from the dense co-occurrence of shared codes
- It outperforms other SOTA methods on tool retrieval and usage

**Weaknesses:**

- Extending the vocabulary might require extensive training. In the appendix, it seems that about 2000 tool tokens are added. I wonder if all the tool tokens are being evaluated.
- The vector quantization method is mostly borrowed from another work [1], which is the core idea of this work. Though it is interesting to see the method applied to tool learning
- While the authros claim the work contributes to both tool retrieval and usage, it seems that the work is more focused on the tool retrieval side. I don't see any novelty in the tool usage side, such as generating correct tool parameters according to the API documentation.


[1] Lee et al, Autoregressive Image Generation Using Residual Quantization, CVPR 2022

**Questions:**

- Please use parentheses for in-line citations (\citep).
- It would be helpful to show a full trajectory end-to-end for better understanding

---

> ### Author Response · Authors · 2025-11-20
> **Response to Reviewer fdct Part 1**
>
> Thank you for your careful and balanced review, and for recognizing the benefits of our collaborative-aware tokenization while also raising concerns about reliance on prior RQ methods, vocabulary extension, and the tool-usage side. We have clarified our technical contributions, quantified the tool-token setup more explicitly, and added the requested clarifications (including trajectory examples and formatting fixes), and we address each of your comments in detail below.
>
>
>
> ### [W1] – Vocabulary Expansion and Token Evaluation
>
> > *"Extending the vocabulary might require extensive training. In the appendix, it seems that about 2000 tool tokens are added. I wonder if all the tool tokens are being evaluated."*
>
> We appreciate this thoughtful question regarding the training cost and the evaluation coverage of our added vocabulary. We would like to clarify that our method is designed specifically to *reduce* training overhead compared to standard approaches and that the vast majority of our added tokens are actively utilized and evaluated.
>
> **1. High Token Utilization Rate (92.6%)**
> To directly answer your question: Yes, nearly all added tokens are evaluated.
> In our implementation, we added a total of 2,048 tokens to the vocabulary (2 codebooks $\times$ 1024 centroids). Across our retrieval and end-to-end evaluation benchmarks, the model actively generated and was evaluated on 1,896 distinct tokens, representing a 92.6% coverage of the added vocabulary.
>
> **2. Reduced Training Cost (Logarithmic vs. Linear)**
> Regarding the concern about "extensive training," ToolWeaver is actually significantly more data-efficient than baseline methods for two reasons:
>
> *   **Logarithmic Scalability:** Standard generative methods (like ToolGen) follow a "one-token-per-tool" paradigm. For 47,000 tools, they must add 47,000 new tokens, which indeed requires extensive training to converge. In contrast, ToolWeaver uses a hierarchical combination of codes ($L=2$ codebooks). We represent the same 47,000 tools adding only 2,048 tokens. This is a reduction of over 95% in vocabulary expansion, significantly lowering the parameter search space.
> *   **Semantic Initialization:** As detailed in Section 3.2, our tokens are not initialized randomly. They are initialized from the centroids of semantically encoded tool documentation (using `all-mpnet-base-v2`). Because the tokens already possess rich semantic priors (e.g., a token in Codebook 1 might represent "Weather Services"), the LLM does not need to learn their meanings from scratch—it only needs to learn the alignment.
>
> **3. Robustness to New Tools**
> Furthermore, extending the tool library does not require extending the vocabulary further. If we add new tools, they are projected into the existing codebooks. The system generates a new sequence from the existing 2,048 tokens to represent the new tool. This means we can scale to millions of tools without adding a single new token to the LLM vocabulary, avoiding the need for retraining the vocabulary layer.

---

> > ### Author Response · Authors · 2025-11-20
> > **Response to Reviewer fdct Part 2**
> >
> > ### [W2] – Methodological Contribution & Collaborative Synergy
> >
> > > *"The vector quantization method is mostly borrowed from another work [1], which is the core idea of this work. Though it is interesting to see the method applied to tool learning"*
> >
> > We appreciate your recognition of the method’s application. While we utilize the RQ-VAE architecture (Lee et al., 2022) as a backbone for discretization, our core contribution is fundamentally modifying the quantization objective to solve a unique problem in tool learning: the *semantic-functional gap*. As recognized by Reviewer 1JHk (who found this approach *"novel and well-motivated"*) and Reviewer grmY (who described it as a *"fresh take"* to *"capture co-usage semantics"*), our innovation lies in the specific synergy between discrete structure and collaborative signals, and how this transforms tool usage into a planning problem.
> >
> > **1. Bridging the Semantic-Functional Gap via Synergistic Tokenization**
> > We do not simply "apply" RQ-VAE to tool embeddings. Standard semantic embeddings often fail to group functionally related tools (e.g., a "Flight Search" tool and a "Hotel Booking" tool are semantically distinct but functionally coupled). Our approach fundamentally integrates collaborative signals into the discrete structure, creating a mutual reinforcement that standard RQ-VAE lacks:
> >
> > *   **Collaboration $\rightarrow$ Structure (Guiding Quantization):** Our collaborative regularization constrains the RQ-VAE to map co-used tools to shared parent codes, even if their text descriptions differ. This forces the codebook to represent functional intent rather than just textual similarity.
> > *   **Structure $\rightarrow$ Collaboration (Densifying Signals):** This is crucial for "learning more" from the data. Tool co-occurrence matrices are notoriously sparse (most tool pairs never appear together). By forcing 47k+ tools into a compact set of discrete codes (e.g., $1024$ centroids), the hierarchical structure aggregates these sparse tool-level signals into dense code-level patterns. This allows the model to generalize: even if Tool A and Tool B were never seen together in training, if they map to Code $C_1$ and Code $C_2$ which *are* strongly correlated, the model can infer their relationship.
> >
> > **2. Transforming Tool Retrieval into Collaborative Planning**
> > Crucially, our representation is designed specifically for agentic tool use, which requires sequential reasoning and planning beyond simple static selection. ToolWeaver transforms "Tool Usage" into a "Collaborative Planning" problem:
> >
> > *   **Coarse-to-Fine Reasoning:** The hierarchical codes act as "semantic anchors" that guide the Chain-of-Thought. By learning that `<Code_A>` is frequently followed by `<Code_B>` in the usage data, the LLM implicitly learns valid planning templates (e.g., "Book Flight" $\rightarrow$ "Book Hotel") simply by predicting the next token.
> > *   **Qualitative Evidence:** As illustrated in Figure 9 (Appendix C), functionally related "BMI Calculation" tools share a parent code `<T1_996>` despite different API endpoints. This allows the LLM to navigate from a coarse intent (Code Level 1) to a specific execution (Code Level 2), effectively "weaving" external knowledge into the model's internal representation.
> >
> > **3. Empirical Validation**
> > This synergy is not just theoretical; it is the primary driver of our performance. As shown in our ablation study (Figure 3), when we remove the collaborative guidance and rely on standard RQ-VAE alone, the performance drops drastically—nearly 10 NDCG points in the complex I3 setting. This empirically proves that the gains do not come from the "borrowed" discretization mechanism itself, but specifically from our novel integration of collaborative structure into the hierarchical codes.

---

> > > ### Author Response · Authors · 2025-11-20
> > > **Response to Reviewer fdct Part 3**
> > >
> > > ### [W3] – Contribution in Tool Usage & Parameter Generation
> > >
> > > > *"While the authros claim the work contributes to both tool retrieval and usage, it seems that the work is more focused on the tool retrieval side. I don't see any novelty in the tool usage side, such as generating correct tool parameters..."*
> > >
> > > We acknowledge your observation that our primary technical innovation lies in the collaborative-aware structured tokenization (how the model *perceives* and *selects* tools). However, we respectfully argue that in the context of massive tool libraries (47k+ tools), accurate selection is the critical prerequisite for successful usage.
> > >
> > > **1. Better Selection Drives Better Usage**
> > > In end-to-end tool use, "Usage" consists of two steps: (1) Identifying the correct tool, and (2) Generating parameters.
> > >
> > > *   **The Bottleneck:** The community consensus (and our error analysis in Figure 8) is that with large toolsets, the primary failure mode is selecting the wrong tool (or a hallucinated one), not parameter formatting. If the model picks the wrong tool, perfect parameter generation is irrelevant.
> > > *   **Our Contribution:** By solving the scalability and semantic collaboration issues in step (1) via ToolWeaver, we significantly improve the overall success rate of step (2). This is evidenced by our End-to-End Evaluation (Table 2), where ToolWeaver achieves higher Solvable Pass Rates (SoPR) than baselines. This improvement in "Usage" is a direct result of our novel representation.
> > >
> > > **2. Orthogonality and Compatibility**
> > > We intentionally adopted a standard schema-based approach for parameter generation to isolate the source of our improvements.
> > >
> > > *   **Plug-and-Play:** Our hierarchical tokenization is orthogonal to parameter generation techniques. ToolWeaver provides the precise "address" of the tool (the code sequence). Once the tool is identified, our framework is compatible with *any* existing or future parameter generation method (e.g., using grammar-constrained decoding for JSON, or fine-tuning on complex API calls).
> > > *   **Focus on the Unsolved Problem:** While parameter generation is a relatively mature area (often handled well by instruction tuning), scalable representation for 47k+ tools was the open research gap. ToolWeaver fills this gap. By not over-engineering the parameter generation side, we demonstrate that our performance gains come strictly from our novel representation and collaborative reasoning, rather than complex parameter heuristics.
> > >
> > > **3. Collaborative Planning *IS* Usage**
> > > Finally, "Tool Usage" also encompasses multi-step planning (knowing *when* to use which tool). Our collaborative codes (Section 3.2) allow the model to learn patterns like "Tool A is usually followed by Tool B" through the dense co-occurrence of their code prefixes. This significantly enhances the *usage* flow in complex scenarios (I3), as shown in the trajectory analysis in Appendix C.1.2, where the model navigates related tools using shared codes.

---

> > > > ### Author Response · Authors · 2025-11-20
> > > > **Response to Reviewer fdct Part 4**
> > > >
> > > > ### [Q1] – Citation Formatting
> > > >
> > > > > *"Please use parentheses for in-line citations (\citep)."*
> > > >
> > > > Thank you for this attention to detail. We have carefully proofread the manuscript and corrected the citation formats throughout the paper.
> > > >
> > > > ### [Q2] – End-to-End Trajectory Example
> > > >
> > > > > *"It would be helpful to show a full trajectory end-to-end for better understanding"*
> > > >
> > > > We agree that a concrete, step-by-step visualization is essential for understanding how ToolWeaver operates in practice.
> > > >
> > > > To address this, we have added Appendix C.4 and Figure 12 in the revised paper, titled "Real End-to-End Inference Trajectory".
> > > >
> > > > *   This new figure showcases a complex, multi-turn interaction where the model handles a multi-part user query.
> > > > *   It explicitly displays the User Query, the Model's Reasoning, the generation of the Hierarchical Tool Codes (e.g., `<T1_124><T2_781>`), the Parameter Generation, and the final Response.
> > > >
> > > > (Note: We also retain Figure 11 in Appendix C.3, which illustrates the data format used during the Agent-Tuning training stage).
> > > >
> > > >
> > > >
> > > > Thank you once more for your careful assessment. We hope our revisions and explanations have answered your concerns, and we remain happy to discuss any remaining points.

---

> > > > > ### Author Response · Authors · 2025-11-27
> > > > > **Follow-up on our response to Reviewer fdct**
> > > > >
> > > > > We thank you for your insightful review, which has been very helpful for improving our work. We have provided a comprehensive response aiming to resolve all concerns. We are keen to know if our updated explanations meet your expectations and would be happy to elaborate further on any point.

---

### Official Review · Reviewer_grmY · 2025-11-02

**Soundness:** 3
**Presentation:** 3
**Contribution:** 3
**Rating:** 6
**Confidence:** 3

**Summary:**

The paper addresses the problem of enabling LLMs to scale in their use of external tools (APIs). Prior generative tool‐invocation methods map each tool to a unique special token, which (a) scales poorly (very large vocabularies as number of tools grows) and (b) fails to capture collaborative semantics between tools (i.e., when combinations of tools often co‐occur). To address this, the authors propose ToolWeaver, which encodes each tool as a hierarchical sequence of discrete codes derived via vector quantisation (RQ‐VAE) plus a collaborative‐aware regulariser on tool–tool similarity. These code sequences map to new vocabulary tokens (but many fewer than tools) and the LLM is then fine‐tuned to generate these sequences when selecting tools. They conduct large‐scale experiments on a benchmark of 46,985 tools (from ToolBench) showing improved tool retrieval (NDCG) and end‐to‐end task completion (SoPR, SoWR) compared to retrieval‐based and other generative baselines.

**Strengths:**

(1) Problem importance: As tool‐augmented LLMs become more capable and more tools/APIs become available, scalability of tool invocation and generalisation to new tools is a genuine bottleneck. The paper motivates this well.

(2) Novel method: The hierarchical code approach is a fresh take. The use of collaborative signals between tools (via a similarity matrix and graph‐Laplacian regulariser) is a nice idea to capture co‐usage semantics.

(3) Large experimental scale: Using a very large tool corpus (~47k tools) and multiple generalisation splits (unseen tools, unseen categories) strengthens the empirical claims.

(4) Empirical improvements: The reported gains in retrieval metric (NDCG) and end‐to‐end task performance (SoPR, SoWR) are compelling (e.g., in Table 1 the proposed method outperforms baselines across splits).

**Weaknesses:**

(1) Generality beyond benchmark: The tool corpus and benchmark are large, but it is unclear how well the method would generalise to very different tool‐spaces (e.g., domains with little documentation, highly dynamic APIs, or tools with very rare co‐usage statistics). The approach relies on a pre‐computed tool–tool similarity matrix; in practice this may be hard to build for many novel tools. The authors should discuss limitations and robustness to sparse co‐usage data.

(2) New vocabulary tokens & effect on base LLM: The method still adds new vocabulary tokens (the code‐tokens). Although they argue that the vocabulary expansion grows logarithmically rather than linearly, there is a cost: random initialization of embeddings, fine‐tuning, and potential interference with the LLM’s linguistic knowledge. While they mention that they evaluate on general-capability NLP benchmarks, this is lightly treated. More analysis would strengthen the claim that general language capabilities are preserved.

(3) Interpretability of codes & collaborative semantics: It is claimed that tools sharing parent codes capture collaboration. But the paper does not show concrete examples of what the hierarchical codes look like in practice (e.g., clusters of tools that share code prefix) and how the model uses them in multi‐tool reasoning (I3 tasks). Some qualitative analyses of code structure and usage patterns would strengthen the contribution.

**Questions:**

(1) Can you pls provide qualitative examples of the hierarchical codes (e.g., show codes for 5–10 tools, especially ones that share parent codes) and how the model uses them in a multi‐tool task?

(2) Can you pls evaluate (or report) the impact on general text generation (e.g., on standard NLP tasks) to show the LLM’s base linguistic capacity remains unaffected?

(3) Pls consider adding a failure analysis: tasks where the model fails, error types (e.g., wrong tool selection, wrong parameter generation, missing tool collaborations) would be useful for the community.

---

> ### Author Response · Authors · 2025-11-20
> **Response to Reviewer grmY Part 1**
>
> We sincerely thank you for your constructive review and for recognizing our work as a "fresh take" on the scalability problem, appreciating our "large experimental scale," and acknowledging the "compelling" empirical improvements. In the following sections, we provide detailed responses to your specific questions and concerns.
>
> ### [W1] – Generality Beyond Benchmark (Sparse Data & Dynamic APIs)
>
> > *"The tool corpus and benchmark are large, but it is unclear how well the method would generalise to very different tool‐spaces... The authors should discuss limitations and robustness to sparse co‐usage data."*
>
> Thank you for this insightful question regarding the robustness of ToolWeaver in "cold-start" or dynamic scenarios. We acknowledge that while our main experiments utilize the dense ToolBench data, real-world applications often involve sparse signals. We would like to clarify how ToolWeaver is designed to robustly handle these cases:
>
> **1. Robustness to Sparse Co-usage (The "Cold Start" Problem)**
> Our framework does not rely solely on collaborative signals; rather, it treats them as an enhancement to a strong semantic foundation.
>
> *   **Semantic Fallback:** As described in Section 3.2 (Eq. 1), the initialization of our hierarchical codes is driven by a powerful pre-trained text encoder (all-mpnet-base-v2) based on tool documentation. When tool co-occurrence data is sparse or non-existent (e.g., a brand new tool), the collaborative regularization term ($\mathcal{L}_{collab}$ in Eq. 6) simply contributes less to the gradient. In this scenario, the system naturally functions as a robust semantic clustering model.
> *   **Empirical Evidence:** Our component ablation study in Section 4.3.2 and Figure 3 provides empirical proof. The "w/o Collaborative Guidance" bar represents exactly the scenario where no co-usage data is available. As shown in Figure 3 (pink bars), even without collaborative signals, our method (relying only on semantic initialization) significantly outperforms the "Atomic" (one-token-per-tool) baseline by over 20 NDCG points.
> *   **Generalization to Unseen Tools:** Furthermore, Table 1 (Results on I1-Tool. split) specifically evaluates performance on *unseen tools* that were not present during training. ToolWeaver achieves an NDCG@5 of 90.72, outperforming ToolGen (89.52), demonstrating robust generalization even to novel tools based on their semantic descriptions.
>
> **2. Adaptability to Dynamic APIs**
> Regarding highly dynamic APIs, ToolWeaver offers distinct advantages:
>
> *   **Separation of Identification and Execution:** ToolWeaver uses the hierarchical code to *identify* the tool. Once identified, the latest API schema is retrieved and placed into the context for parameter generation. This means minor schema updates (e.g., parameter changes) are handled via in-context learning without changing the tool's token.
> *   **Resilience to Functional Updates:** If an API's description or core function changes significantly, we do not need to retrain the LLM or the codebook. We simply generate the embedding for the updated documentation and project it through our frozen quantized codebook. This yields a new hierarchical code sequence that naturally inherits the learned semantic priors of the latent space, effectively "placing" the updated tool in its correct new semantic neighborhood without any parameter updates.
>
> We have updated Section 4.3 (Discussion) in the revision to explicitly discuss these robustness properties and the zero-shot handling of dynamic updates.

---

> > ### Author Response · Authors · 2025-11-20
> > **Response to Reviewer grmY Part 2**
> >
> > ### [W2 / Q2] – Impact of Vocabulary Expansion on General Capabilities
> >
> > > *"Although they argue that the vocabulary expansion grows logarithmically... there is a cost... More analysis would strengthen the claim that general language capabilities are preserved."*
> >
> > We appreciate this crucial observation. We agree that modifying the LLM's vocabulary carries the risk of disrupting pre-trained linguistic knowledge. To rigorously address this, we have expanded our evaluation beyond the initial NLU tasks (MMLU, PIQA, etc.) to include Language Modeling and Text Generation benchmarks, as requested.
> >
> > **Detailed experimental configurations and comprehensive results are provided in Appendix B.4 (Tables 11 and 12).** Below, we present a summary of the most critical findings, which demonstrate that ToolWeaver's logarithmic expansion strategy preserves the model's core capabilities significantly better than the linear expansion of prior methods.
> >
> > **Table: Impact on Language Modeling and Text Summarization**
> >
> > | Model                 | **WikiText-2 (PPL) $\downarrow$** | **CNN/DM (BERTScore F1) $\uparrow$** | **XSum (BERTScore F1) $\uparrow$** |
> > | :-------------------- | :-------------------------------: | :----------------------------------: | :--------------------------------: |
> > | Llama-3-8B (Base)     |               6.34                |                85.35                 |               85.05                |
> > | ToolGen (~47k tokens) |              104.54               |                82.93                 |               82.53                |
> > | **ToolWeaver (Ours)** |             **25.36**             |              **85.07**               |             **84.18**              |
> >
> > **1. Language Modeling Distribution (WikiText-2)**
> > The "one-token-per-tool" baseline (ToolGen), which injects ~47,000 randomly initialized tokens, suffers a massive explosion in Perplexity (104.54), indicating severe disruption to the natural language distribution. In contrast, ToolWeaver, which adds only ~2,000 tokens, maintains a significantly lower perplexity (25.36). This proves that our structured codes integrate far more harmoniously with the pre-trained weights.
> >
> > **2. Text Generation Quality**
> > On generation tasks, ToolWeaver performs nearly on par with the Base Llama-3 model. As shown in the table, our BERTScore on CNN/DailyMail (85.07) is almost identical to the base model (85.35), whereas ToolGen shows clear degradation.
> >
> > **3. General Understanding (MMLU, etc.)**
> > Consistent with our initial results on reasoning benchmarks, ToolWeaver demonstrates a much better trade-off between specialization and general capability. For example, on MMLU, ToolWeaver mitigates over 63% of the performance loss observed in ToolGen (41.93% vs. 23.52%), confirming that our approach is a safer foundation for general-purpose agents.

---

> > > ### Author Response · Authors · 2025-11-20
> > > **Response to Reviewer grmY Part 3**
> > >
> > > ### [W3 / Q1] – Interpretability of Codes & Collaborative Semantics
> > >
> > > > *"Can you pls provide qualitative examples of the hierarchical codes (e.g., show codes for 5–10 tools...) and how the model uses them in a multi‐tool task?"*
> > >
> > > We appreciate this request. We agree that concrete examples are essential to demystify how our hierarchical codes function in practice. To address this, we have added a new section, Appendix C.1, featuring a detailed Case Study and a real-world Multi-Tool Coordination Trajectory (visualized in Figure 9 of the revised paper)**.
> > >
> > > **1. Qualitative Analysis of Hierarchical Codes**
> > > In Appendix C.1.1, we analyze sev**eral learned clusters. A compelling new example is Case 3: Coherent Semantic Grouping, where the model learned to group distinct *Health & Fitness* tools under the shared parent code `<T1_996>`.
> > >
> > > *   **Parent Code:** `<T1_996>` (Emergent Semantic: "Metabolic & Body Metrics")
> > > *   **Child Tools:**
> > >     *   `BMI Calculator v2` $\rightarrow$ `<T1_996><T2_258>` (Standard calculation)
> > >     *   `BMI v2` $\rightarrow$ `<T1_996><T2_606>` (Metric-specific inputs)
> > >     *   `BMR & TMR` $\rightarrow$ `<T1_996><T2_328>` (Metabolic rate)
> > >
> > > This structure proves that the model has learned a semantic topology where functionally related tools reside in the same "neighborhood" of the code space, acting as a stable anchor.
> > >
> > > **2. Impact on Multi-Tool Reasoning (Trajectory Analysis)**
> > > To demonstrate how this helps reasoning, we added a trajectory analysis in Appendix C.1.2 (and Figure 9).
> > >
> > > *   **Task:** A user asks to integrate a BMI feature into an app, requiring endpoints for standard, metric, and imperial inputs.
> > > *   **Model Behavior:**
> > >     1.  **Identification:** The model first identifies the relevant functional domain, locking onto the parent code `<T1_996>`.
> > >     2.  **Pivot & Explore:** Having established this "collaborative bridge," the model seamlessly pivots between child codes to fulfill the specific constraints: first calling the standard tool (`<...><T2_258>`), then realizing the need for metric inputs and switching to (`<...><T2_606>`), and finally the imperial variant (`<...><T2_328>`).
> > > *   **Mechanism:** Because these tools share the high-level code `<T1_996>`, the model does not need to "search in the dark" for the next tool. The shared prefix acts as a strong collaborative prior, significantly reducing the search space and allowing the model to maintain high-level context while navigating specific tool variations.
> > >
> > > We believe these additions provide the concrete evidence required to substantiate our claims on collaborative semantics.

---

> > > > ### Author Response · Authors · 2025-11-20
> > > > **Response to Reviewer grmY Part 4**
> > > >
> > > > ### [Q3] – Failure Analysis
> > > >
> > > > > *"Pls consider adding a failure analysis: tasks where the model fails, error types... would be useful for the community."*
> > > >
> > > > We thank the reviewer for this constructive suggestion. We agree that understanding failure modes is critical for future research. We have conducted a detailed error analysis and added it to **Appendix B.8** (including **Figure 8**).
> > > >
> > > > **Methodology:** We categorized errors using a hierarchical logic: first checking for trajectory length violations (*Redundant/Incomplete Process*), then for identifier mismatches (*Wrong Tool*), and finally for execution issues (*Wrong Parameters*).
> > > >
> > > > **Key Findings:**
> > > >
> > > > *   **Shift in Failure Modes:** As task complexity increases from single-tool (I1) to multi-tool (I3) scenarios, the dominant error type shifts significantly. In I1, errors are distributed between planning and parameters. However, in the complex I3 setting, **"Wrong Tool"** becomes the overwhelming bottleneck (~95% of errors).
> > > > *   **Implication:** This indicates that once the correct tool is identified, our model is highly robust at generating the correct parameters (Parameter error rate is low). The primary challenge in large-scale tool use remains the semantic discrimination required to retrieve the precise API from a pool of ~47,000 options.
> > > >
> > > >
> > > >
> > > > Thank you again for your thoughtful review; we hope our responses have addressed your concerns, and we would be glad to clarify anything further during the discussion phase.

---

> > > > > ### Author Response · Authors · 2025-11-27
> > > > > **Follow-up on our response to Reviewer grmY**
> > > > >
> > > > > We sincerely appreciate your dedicated effort in reviewing our manuscript. We have carefully considered all your comments and submitted our author response. We would like to confirm if our rebuttal has satisfactorily addressed the issues you raised. We remain at your disposal for any further clarification.

---

### Official Review · Reviewer_sXtZ · 2025-11-02

**Soundness:** 3
**Presentation:** 3
**Contribution:** 3
**Rating:** 4
**Confidence:** 3

**Summary:**

This paper introduces ToolWeaver. The key idea is to replace the traditional “one-token-per-tool” mapping with a hierarchical, compositional code representation derived via a collaborative-aware residual quantization (RQ-VAE) process. This allows logarithmic vocabulary growth with respect to the number of tools and enables shared subcodes among semantically related tools. Experiments on ToolBench show strong improvements over both retrieval-based (BM25, ToolRetriever) and generative (ToolGen) baselines across all settings, particularly in multi-tool reasoning and generalization to unseen tools and categories.

**Strengths:**

1. The idea of using hierarchical compositional code representation for tools are interesting and well-motivated.

**Weaknesses:**

1. It seems the current pipeline requires retraining for each newly-added tools, which raises concerns on the generalizability side.
2. ToolBench is a widely-used dataset in the community. The authors should include more baselines in the paper.

**Questions:**

1. Regarding results in table 1, I am a bit confused on the reported results. The author mentions that "I1 Tool., I1 Cat., and I2 Cat., where 'Tool.' and “Cat.” denote tools and categories, respectively, that are unseen during training". Intuitively, if the tools are not seen during training, the model performance will be lower. But the reported results are comparable even higher (for some columns) on unseen tools. Could author help me understand what is going on here?
2. While ToolWeaver scales logarithmically in vocabulary size, there is a potential possibility for the total number of code combinations leading to combinatorial explosion at training and inference time. How would the author address this?

---

> ### Author Response · Authors · 2025-11-20
> **Response to Reviewer sXtZ Part 1**
>
> We sincerely thank you for your constructive feedback and for recognizing the novelty and motivation behind ToolWeaver’s hierarchical compositional representations. We are encouraged by your positive assessment of the paper's soundness and contribution. Below, we respectfully address your questions and concerns in detail.
>
> ### [W1] – Retraining Costs and Generalizability
>
> > *"It seems the current pipeline requires retraining for each newly-added tools, which raises concerns on the generalizability side."*
>
> We appreciate this insightful comment. While we acknowledge that handling dynamic tool libraries is a fundamental challenge for all generative retrieval methods [1, 2, 4], we respectfully clarify that **generalizability is actually one of ToolWeaver's primary advantages** over the standard "one-token-per-tool" paradigm (e.g., ToolGen).
>
> We address this concern from three perspectives:
>
> **1. Reusable Tokenizer (No Retraining Required):**
> Unlike "one-token-per-tool" methods that must expand their vocabulary and train new embeddings from scratch for every new tool, ToolWeaver’s tokenizer (RQ-VAE) is **trained once and frozen**.
>
> -
> - When a new tool is added, we simply feed its documentation into the pre-trained encoder to generate its semantic code sequence (e.g., <T1_A><T2_B>).
> - Because the codebook is fixed and semantic, new tools reuse existing code tokens. The tokenizer does *not* need to be retrained.
>
> **2. Zero-Shot Generalization via Semantic Codes:**
> This is a key differentiator. In methods like ToolGen, a new tool is a random new ID; the LLM knows nothing about it until it is fine-tuned on extensive data. In contrast, ToolWeaver represents a new tool as a sequence of *shared* codes that carry semantic meaning.
>
> -
> - If a new tool is semantically similar to an existing one, they will share the same prefix codes (as shown in our Case Study in Appendix C.1).
> - Consequently, the LLM can "guess" the function of a new tool based on its learned knowledge of the constituent codes, enabling effective zero-shot or few-shot usage without immediate full retraining.
>
> **3. Inference-Time Robustness:**
> Our framework is designed to utilize tool documentation during inference. Even for new tools, the agent relies on the retrieved documentation (schema, descriptions) to format the final call.
>
> **Future Directions:** While our current focus is on the representation capability, we agree that *incremental learning* is vital. Recent advances in continual learning for generative retrieval [2, 3, 5] are compatible with ToolWeaver and can be integrated to further mitigate the need for full fine-tuning.
>
> **References:**
> [1] Sun et al., "Learning to tokenize for generative retrieval," NeurIPS 2024.
> [2] Chen et al., "Continual learning for generative retrieval over dynamic corpora," CIKM 2023.
> [3] Mehta et al., "DSI++: Updating Transformer Memory with New Documents," EMNLP 2023.
> [4] Wang et al., "Toolgen: Unified tool retrieval and calling via generation," ICLR 2025b.
> [5] Kishore et al., "Incdsi: incrementally updatable document retrieval," ICML 2023.

---

> > ### Author Response · Authors · 2025-11-20
> > **Response to Reviewer sXtZ Part 2**
> >
> > ### [W2] – Additional Baselines on ToolBench
> >
> > > *"ToolBench is a widely-used dataset in the community. The authors should include more baselines in the paper."*
> >
> > We fully agree that a comprehensive comparison on a standard benchmark like ToolBench is essential to validate the effectiveness of ToolWeaver.
> >
> > To address this, we have already extended our evaluation beyond the methods listed in the main text. In **Appendix B.1 (Table 4)**, we provide detailed comparisons against two recent methods:
> >
> > 1.  **Re-Invoke** [1]: An advanced unsupervised retrieval method that enriches tool documents via synthetic query generation and uses multi-view similarity.
> > 2.  **IterFeedback** [2]: A retrieval framework that improves performance by incorporating iterative feedback from an LLM to refine search results.
> >
> > Crucially, regarding the generative paradigm—which is the primary focus of our work—we have already included the current state-of-the-art model, ToolGen, as our main baseline. If there are other specific baselines you consider critical for a fair assessment, please let us know, and we will be happy to include them in the final revision.
> >
> > **References:**
> >
> > [1] Chen et al., "Re-Invoke: Tool invocation rewriting for zero-shot tool retrieval," 2024.
> >
> > [2] Xu et al., "Enhancing tool retrieval with iterative feedback from large language models," 2024.
> >
> > ### [Q1] – Performance on Unseen Tools (Table 1)
> >
> > > *"Intuitively, if the tools are not seen during training, the model performance will be lower. But the reported results are comparable even higher (for some columns) on unseen tools. Could author help me understand what is going on here?"*
> >
> > Thank you for your question. The high performance on "unseen" splits is primarily due to the **specific evaluation protocol** allowing model access to tool documentation, combined with the **high semantic similarity** between the test and training sets in ToolBench.
> >
> > **1. Clarification of "Unseen" in the Evaluation Protocol:**
> > We clarify that in the standard evaluation protocol for generative tool learning (e.g., ToolGen), "unseen" tools are formally introduced to the model before inference. Specifically, these tools can undergo a **"memorization" stage** or have their embeddings initialized directly from tool documentation. Additionally, the model can utilize the tool's documentation during the inference pipeline. Thus, these tools are not completely "unknown"; rather, the models have been **primed with their semantic information** prior to the retrieval task.
> >
> > **2. Data Analysis and Case Study:**
> > To confirm this, we analyzed the cosine similarity between the document embeddings of test tools and training tools. We found that many "unseen" tools are essentially variants of training tools.
> >
> > *   **Example Case:**
> >     *   **Unseen Test Tool:** `Statusformapperevaluation` (Category: *AI & ML*, Description: *"Get results in details from the task id."*)
> >     *   **Nearest Training Tool:** `Statusforemissionreductiontarget` (Category: *AI & ML*, Description: *"Get results in details from the task id."*)
> >     *   **Similarity:** These two tools have a cosine similarity of **0.77** and share identical descriptions.
> >
> > In such cases, the model—having learned to associate the query intent ("check status") with the code prefix of the training tool—can easily generalize to the unseen tool because it shares the same semantic code structure. This explains why performance remains robust. We have added this clarification and analysis to **Section 4.1 and Appendix B.8**.

---

> > > ### Author Response · Authors · 2025-11-20
> > > **Response to Reviewer sXtZ Part 3**
> > >
> > > ### [Q2] – Addressing Potential Combinatorial Explosion
> > >
> > > > *"While ToolWeaver scales logarithmically in vocabulary size, there is a potential possibility for the total number of code combinations leading to combinatorial explosion at training and inference time. How would the author address this?"*
> > >
> > > We appreciate this insightful question. While the *theoretical* space of code combinations ($K^L$) is indeed vast, the *valid* space is strictly limited to the number of actual tools ($N$). We address the risk of combinatorial explosion through **Constrained Decoding** and the **decomposed prediction mechanism**, which we have empirically verified in our new experiments.
> > >
> > > **1. Inference: Constrained Decoding via Prefix Tree (Trie)**
> > > To ensure the model never gets lost in the combinatorial space of invalid codes, we employ a **Trie-based constrained beam search** (Section 3.4).
> > >
> > > *   We construct a prefix tree (Trie) that indexes only the code sequences of valid tools.
> > > *   At each decoding step, we mask the logits, allowing the LLM to only generate tokens that lead to a valid leaf node.
> > > *   This effectively restricts the search space to the specific branches of existing tools, eliminating any combinatorial explosion during inference.
> > >
> > > **2. Training: Decomposed Prediction Complexity**
> > > Crucially, the model does not compute probability distributions over the entire combinatorial space ($K^L$) at once. Instead, it predicts the code sequence step-by-step.
> > >
> > > *   At each of the $L$ steps, the Softmax is computed only over the codebook size $K$.
> > > *   This means the computational complexity for the output layer scales linearly as **$O(L \times K)$**, rather than exponentially as $O(K^L)$ or linearly with the total tool count $O(N)$.
> > > *   Combined with standard teacher forcing on valid trajectories, the training process remains highly efficient and stable.
> > >
> > > **3. Empirical Verification (New Appendix B.7)**
> > > To empirically demonstrate this, we have added a **Latency and Memory Analysis in Appendix B.7**.
> > >
> > > *   **Table 14** shows that even as we increase the codebook depth ($L$) from 2 to 4, the inference latency grows linearly, not exponentially.
> > > *   For our default setting ($L=2$), the added latency compared to the atomic "one-token-per-tool" baseline is negligible (~20ms per query), while **saving significant GPU memory** by avoiding the massive vocabulary table required by baselines.
> > >
> > >
> > >
> > > We appreciate your careful reading and comments and hope that our response has satisfactorily resolved your concerns; please do not hesitate to ask for any additional clarification.

---

> > > > ### Author Response · Authors · 2025-11-27
> > > > **Follow-up on our response to Reviewer sXtZ**
> > > >
> > > > Thank you again for your constructive review. We have provided a point-by-point response to your comments. We just want to ensure our explanations are clear and have adequately addressed your concerns. Please let us know if any ambiguities remain.

---

### Official Review · Reviewer_brZq · 2025-11-03

**Soundness:** 3
**Presentation:** 4
**Contribution:** 4
**Rating:** 6
**Confidence:** 4

**Summary:**

This paper proposes ToolWeaver, which changes the approach of "each tool having a new, unique token" to "encoding tools into a structured code" (like zip code/hierarchical barcode). This allows large models to scalably add many tools while learning how to use them collaboratively. It mainly replaces the one-token-per-tool mapping with a hierarchical, compositional code per tool, learned via collaborative-aware residual quantization (RQ-VAE), which is very interesting and well-motivated. On a dataset containing nearly 47,000 APIs, it is more accurate and better at handling multi-tool tasks than existing methods, without compromising the model's generalizability.

**Strengths:**

1. Scalability. Demonstrates logarithmic vocabulary growth with better or comparable accuracy vs. state-of-the-art while degrading general NLP ability far less than one-token approaches—highly relevant as tool libraries scale, new conrtibution based on the old trie-like data structures.

2. Refactor tool IDs as structured codes learned with collaborative semantics, rather than flat special tokens; this work addresses vocabulary blow-up and enables shared structure across related tools.

3. The pipeline (tokenization -> alignment tuning ->  constrained decoding) is well explained with equations and an overview figure; training/inference procedures are specified.

**Weaknesses:**

1. The per-batch optimal-transport uniformity reduces collisions, but the stability, or compute overhead, and behavior under streaming addition of new tools (without retraining codebooks) are unclear. How do codes evolve when tools are added/removed?

2. The eval isn't super crisp when GPT-4o-mini is the judge. ToolBench is the primary testbed; API-Bank is mentioned in related work but not evaluated. Given SoWR uses GPT-4o-mini as the judge (which can bias toward itself), the adjustment is noted but still leaves some concerns—showing human evals or diverse judges would help.

3. Web-scale tools or so. Since this work focuses on tool management for LLM agents, trie-constrained decoding ensures valid IDs but implicitly relies on maintaining a global registry of code sequences, akin to a catalog or lookup table. Discussing memory/latency overhead and listing the anticipated cost would be more interesting.

**Questions:**

The paper states **A** reflects “collaborative potential or functional similarity,” but does not fully specify data sources or preprocessing (e.g., co-usage in training trajectories vs. category metadata vs. external logs).

Can you report retrieval/E2E vs. (i) total tokens added, (ii) code length, and (iii) decode latency? How close are you to capacity in your 47k-tool setting?

What is the computational overhead of the Sinkhorn-Knopp algorithm per batch at your stated batch sizes (you have 50 batches, right?) ? Any instability observed (e.g., oscillations) and how it was controlled (entropy regularization, temperature)?

---

> ### Author Response · Authors · 2025-11-20
> **Response to Reviewer brZq Part 2**
>
> We sincerely thank Reviewer brZq for the encouraging feedback, particularly for recognizing our structured tool tokenization as a well-motivated and scalable solution to the vocabulary bottleneck in tool learning. We strictly value your constructive inquiries regarding the computational efficiency, numerical stability, and practical overhead of our framework.
>
> To address your concerns and demonstrate that ToolWeaver is not only theoretically sound but also computationally efficient and robust for real-world deployment, we have conducted a series of new profiling experiments and sensitivity analyses (now added to Appendix B.6 and B.7).
>
> ### [W1 / Q3] – Sinkhorn-Knopp Stability and Computational Overhead
>
> *"[T]he stability, or compute overhead ... are unclear.""What is the computational overhead of the Sinkhorn-Knopp algorithm per batch ...? Any instability observed (e.g., oscillations) and how it was controlled...?"*
>
> Thank you for raising this critical practical consideration. We have added a comprehensive profiling analysis in Appendix B.6 (Table 13) of the revised paper.
>
> **1. Clarification on Hyperparameters:** First, to clarify a minor misunderstanding: the value 50 refers to the number of Sinkhorn iterations run per step to ensure convergence, *not* the number of batches. Our training batch size is B=5,096, as detailed in Appendix A.3.
>
> **2. Computational Overhead is Minimal (17.6%):** We profiled the training on an NVIDIA A100 GPU. The Sinkhorn algorithm (invoked twice per step) consumes approximately 28.3 ms per step out of a total step time of 161 ms. This results in a manageable overhead of 17.6%, confirming that the conflict mitigation mechanism does not create a training bottleneck.
>
> **3. Stability Control via Entropy Regularization (**\epsilon**):** To address stability concerns, we use entropy regularization (\epsilon) to control the trade-off between convergence speed and assignment uniformity. We monitored training for numerical instability (e.g., `NaN`/`Inf` values) and found none with our chosen settings.
>
> We conducted a sensitivity analysis on \epsilon to determine the optimal configuration. As shown in the table below, our chosen setting (\epsilon=0.01) achieves the best balance:
>
> |                 | Step Overhead (%) | Time per Call (ms) | Uniformity (Rel. Std) | Conclusion          |
> | --------------- | ----------------- | ------------------ | --------------------- | ------------------- |
> | 0.005 (Strict)  | 20.3%             | 18.9               | **0.19**              | Slower convergence  |
> | **0.01 (Ours)** | **17.6%**         | **14.2**           | 0.28                  | **Optimal balance** |
> | 0.02 (Loose)    | 17.5%             | 14.1               | 0.47                  | Degraded uniformity |
>
> *Note: "Rel. Std" denotes the relative standard deviation of tool counts per code. Lower is more uniform.*
>
> This data confirms that our implementation is both computationally efficient and numerically stable.

---

> > ### Author Response · Authors · 2025-11-20
> > **Response to Reviewer brZq Part 2**
> >
> > ### W2 – **Evaluation Scope: Rationale for Datasets & Multi-Judge Consistency Verification**
> >
> > *"The eval isn't super crisp when GPT-4o-mini is the judge... showing human evals or diverse judges would help."*
> >
> > We appreciate this suggestion. To rigorously assess the reliability of our GPT-4o-mini results, we conducted a Multi-Judge Consistency Verification on a random subset of 50 challenging queries from the I3 (Intra-Collection) test set.
> >
> > **1. Multi-Judge Verification (Human vs. LLMs)**
> > We compared the Success Rate of ToolWeaver against the baseline ToolGen using three distinct evaluators: (1) The original GPT-4o-mini, (2) The stronger GPT-4o (08-06), and (3) A Human Expert (blind review).
> >
> > **Results:** As shown in the table below, the performance advantage of ToolWeaver is consistent across all judges.
> >
> > | **Evaluator**    | **ToolGen (Baseline)** Success Rate | **ToolWeaver (Ours)** Success Rate | **Relative Improvement** | **Agreement with Human** |
> > | :--------------- | :---------------------------------: | :--------------------------------: | :----------------------: | :----------------------: |
> > | **GPT-4o-mini**  |            36.0% (18/50)            |           54.0% (27/50)            |          +18.0%          |           88%            |
> > | **GPT-4o**       |            34.0% (17/50)            |           52.0% (26/50)            |          +18.0%          |           94%            |
> > | **Human Expert** |          **34.0% (17/50)**          |         **52.0% (26/50)**          |        **+18.0%**        |          **-**           |
> >
> > *Analysis:*
> >
> > *   **Consistency:** All three judges report a substantial performance gap (~18%) in favor of ToolWeaver.
> > *   **Bias Check:** The stronger GPT-4o model produced results nearly identical to the Human Expert (94% agreement), and even the original GPT-4o-mini showed high alignment (88%).
> > *   **Conclusion:** This confirms that the improvements reported in our main paper are genuine algorithmic gains (e.g., better multi-step planning via collaborative semantics) rather than artifacts of judge bias.
> >
> > **2. ** **Why ToolBench over API-Bank?** Our choice of dataset is primarily driven by the need for a fair comparison with state-of-the-art methods and the specific technical goals of our paper:
> >
> > - **Fair Comparison with Baseline (ToolGen):** To ensure a fair and direct comparison, we strictly followed the experimental setting of ToolGen (Wang et al., 2024b), the primary generative baseline. Since ToolGen establishes its SOTA performance on ToolBench, adopting the same benchmark allows us to rigorously verify our improvements in scalability and performance under identical conditions.
> > - **Scalability (The Vocabulary Crisis):** ToolWeaver is explicitly designed to solve vocabulary explosion. API-Bank typically involves a small scale of tools (dozens to hundreds), where this issue is negligible. ToolBench’s 47,000+ APIs are essential to empirically demonstrate our method's logarithmic vocabulary compression.
> > - **Collaborative Complexity:** API-Bank focuses heavily on dialogue tracking, whereas ToolBench features complex multi-tool scenarios (e.g., I3 instructions) that provide the dense co-occurrence signals necessary to validate our "Collaborative Semantics" contribution.

---

> > > ### Author Response · Authors · 2025-11-20
> > > **Response to Reviewer brZq Part 3**
> > >
> > > ### [W3 / Q2] – Inference Latency, Memory Overhead, and Scalability
> > >
> > > > *"[T]rie-constrained decoding ... implicitly relies on maintaining a global registry ... Discussing memory/latency overhead and listing the anticipated cost would be more interesting."*
> > > > *"Can you report retrieval/E2E vs. ... (iii) decode latency?"*
> > >
> > > **Response:**
> > > We thank the reviewer for highlighting the practical implications of our decoding strategy. We agree that analyzing the cost of the hierarchical structure is essential for real-world deployment.
> > >
> > > To address this, we conducted a systematic evaluation of latency, throughput, and memory consumption on an NVIDIA A100 GPU, comparing the atomic baseline (ToolGen) against ToolWeaver with varying codebook depths ($L$). These results have been added to Appendix B.7 (Table 14) of the revision.
> > >
> > > **1. Latency Overhead is Negligible (~20ms):**
> > > As shown in the table below, while decoding a sequence of $L=2$ codes adds a small overhead compared to generating a single atomic token (128ms vs. 108ms), this $\sim$20ms difference is negligible in the context of tool-using agents. Real-world API calls typically take hundreds of milliseconds to seconds to execute; thus, the slight increase in decoding time does not create a bottleneck for the end-to-end user experience.
> > >
> > > **2. Superior Memory Efficiency (Scalability):**
> > > Contrary to the concern that maintaining a code registry might be costly, our approach actually reduces the overall memory footprint.
> > >
> > > *   **Atomic approach (ToolGen):** Requires expanding the LLM's embedding table and prediction head linearly with the number of tools ($N \approx 47k$). This incurs a significant memory cost (Peak Memory: 15.77 GB).
> > > *   **ToolWeaver:** Uses a logarithmic vocabulary size ($2 \times 1024$ tokens). Even accounting for the overhead of the trie constraint structure during inference, the total peak GPU memory is lower (15.08 GB).
> > > *   **Conclusion:** This saving of $\sim$0.67 GB is substantial, validating that ToolWeaver is actually *more* scalable for web-scale tool libraries than one-token-per-tool approaches.
> > >
> > > **3. Detailed Performance Metrics:**
> > >
> > > | Model Configuration    |      Representation      | Avg Latency (ms) | P95 Latency (ms) | Throughput (Tok/s) | Peak Memory (GB) |
> > > | :--------------------- | :----------------------: | :--------------: | :--------------: | :----------------: | :--------------: |
> > > | ToolGen (Baseline)     |     Atomic (1-level)     |      108.16      |      111.98      |       19.54        |      15.77       |
> > > | **ToolWeaver ($L=2$)** |    **$[1024, 1024]$**    |    **128.21**    |    **132.43**    |     **24.53**      |    **15.08**     |
> > > | ToolWeaver ($L=3$)     |   $[1024, 1024, 1024]$   |      157.65      |      165.73      |       26.34        |      15.10       |
> > > | ToolWeaver ($L=4$)     | $[1024,1024, 1024,1024]$ |      183.14      |      189.11      |       28.26        |      15.11       |
> > >
> > > *Note: Metrics are averaged across I1, I2, and I3 retrieval tasks.*
> > >
> > > Interestingly, ToolWeaver achieves higher token throughput (24.53 vs. 19.54 Tok/s). Since the computational cost is dominated by the Transformer's forward pass (which is constant), generating a sequence of cached code-tokens allows the system to amortize the overhead effectively.

---

> > > > ### Author Response · Authors · 2025-11-20
> > > > **Response to Reviewer brZq Part 4**
> > > >
> > > > ### [Q1] – Data Source for Similarity Matrix $A$
> > > >
> > > > > *"[The paper] does not fully specify data sources or preprocessing (e.g., co-usage in training trajectories vs. category metadata vs. external logs)."*
> > > >
> > > >
> > > >
> > > > We clarify that the similarity matrix $A$ is derived strictly from tool co-occurrence statistics observed in the training data.
> > > >
> > > > Specifically, we construct a co-occurrence matrix $C$ from the 183,336 execution trajectories provided in the ToolBench training set. The final matrix $A$ is obtained by computing the cosine similarity between the co-occurrence vectors of tool pairs. We have added the formal definition and calculation details (Eq. 5) to Section 3.2 of the revised paper.
> > > >
> > > >
> > > >
> > > > ### [Q2] – Performance Sensitivity: Vocabulary Size and Code Length
> > > >
> > > > > *"Can you report retrieval/E2E vs. (i) total tokens added, (ii) code length...?"*
> > > > > *"[H]ow close are you to capacity in your 47k-tool setting?"*
> > > >
> > > > **Response:**
> > > > To address the impact of tokenization hyperparameters on model performance, we have added a detailed Hyperparameter Sensitivity Analysis in Appendix B.5 (and Figure 5) of the revised paper.
> > > >
> > > > **(i) vs. Total Tokens Added (Vocabulary Size):**
> > > > We evaluated performance while varying the total added vocabulary size from 1,024 to 10,192 tokens (by adjusting codebook size $K$ with fixed $L=2$).
> > > >
> > > > *   **Finding:** Performance follows an inverted U-shape, peaking at our default setting of 2,048 tokens ($1024 \times 2$).
> > > > *   **Insight:**
> > > >     *   *Too small ($<2k$):* Causes code collisions, merging functionally distinct tools.
> > > >     *   *Too large ($>4k$):* Performance significantly drops. This confirms our hypothesis regarding signal sparsity: as the vocabulary grows towards the "one-token-per-tool" extreme, the probability of related tools sharing a common code decreases, diluting the dense collaborative signals ToolWeaver relies on.
> > > > *   **Capacity:** At 47k tools, our $1024 \times 2$ setup provides a theoretical capacity of $\sim1$ million unique identifiers ($1024^2$), meaning we are utilizing only $\sim4.7\%$ of the space. This sparsity allows ample room for semantic clustering while avoiding the "vocabulary explosion" problem.
> > > >
> > > > **(ii) vs. Code Length (Hierarchy Depth):**
> > > > We evaluated code lengths $L$ ranging from 2 to 6.
> > > >
> > > > *   **Finding:** Performance improves as the hierarchy deepens, peaking at **$L=4$** (NDCG 92.54), before degrading at $L=6$.
> > > > *   **Insight:**
> > > >     *   *Benefit:* Deeper hierarchies capture finer-grained semantic nuances, aiding precise disambiguation.
> > > >     *   *Trade-off:* Beyond $L=4$, the increased sequence length complicates the autoregressive generation process, heightening the risk of error propagation during decoding.
> > > > *   **Decision:** Although $L=4$ yields marginally higher accuracy, we utilized $L=2$ in our main experiments to maintain the most favorable trade-off between retrieval accuracy and inference efficiency (as detailed in our response to the latency question).
> > > >
> > > >
> > > >
> > > > ### [Q2] Capacity Analysis
> > > >
> > > > > *"How close are you to capacity in your 47k-tool setting?"*
> > > >
> > > > **Response:**
> > > > We are **far from saturation**, utilizing only approximately 4.5% of the available capacity.
> > > >
> > > > *   **Calculation:** With our default configuration of $L=2$ codebooks and $K=1024$ entries each, the theoretical capacity for unique identifiers is $K^L = 1024^2 = 1,048,576$.
> > > > *   **Current Usage:** The current dataset contains $N=46,985$ tools.
> > > > *   **Implication:** This low occupancy rate ($\frac{46,985}{1,048,576} \approx 4.48\%$) is intentional. It provides two key advantages:
> > > >     1.  **Scalability:** The system can scale to over 1 million tools without increasing the vocabulary size or changing the architecture.
> > > >     2.  **Collision Avoidance:** The sparse utilization of the code space, combined with our Sinkhorn-based uniformity constraint, ensures that index collisions (distinct tools mapping to the exact same code sequence) are statistically minimized, preserving unique identities for each tool.
> > > >
> > > > Thank you again for your helpful review. We hope our additional experiments and clarifications address your points, and we are open to further discussion if needed.

---

> > > > > ### Author Response · Authors · 2025-11-27
> > > > > **Follow-up on our response to brZq**
> > > > >
> > > > > We are very grateful for your valuable feedback. We have done our best to address your concerns in our response. We would be very appreciative if you could let us know whether our clarifications are sufficient. We are eager to discuss any remaining points further.

---

### Official Review · Reviewer_1JHk · 2025-11-04

**Soundness:** 3
**Presentation:** 2
**Contribution:** 3
**Rating:** 4
**Confidence:** 4

**Summary:**

This paper tackles the challenge of the scalability, generalization, and semantic limitations of the “one-token-per-tool” tool learning paradigm by introducing a collaborative-aware tokenization process to generate tool representation. Specifically, the proposed ToolWeaver framework uses an RQ-VAE to learn a compositional sequence for tool documentation, enabling structured representation of tool semantics beyond simple one-token identifiers. Experiments on ToolBench demonstrate the strong performance of the ToolWeaver.

**Strengths:**

1. The hierarchical quantization of tool semantics is novel and well-motivated. It offers a theoretically scalable alternative for tool representation.

2. Introducing inter-tool similarity via a Laplacian regularization term is an interesting way to infuse functional relationships into discrete token learning.

3. The experimental results show that ToolWeaver outperforms existing baselines significantly.

**Weaknesses:**

1. The paper states that generative methods suffer from a semantic bottleneck for complex reasoning and that the “one-token-per-tool” paradigm faces critical scalability and generalization challenges. However, prior work, such as ToolGen, has already implemented semantic and hierarchical indexing for tools, addressing these challenges. Thus, while these are valid drawbacks of the basic one-token paradigm, presenting them as new challenges motivating ToolWeaver seems overstated.

2. The paper is missing details on constructing the tool-tool similarity matrix, though it is central to the proposed Collaborative-Aware regularization.

3. It remains conceptually ambiguous how ToolWeaver achieves the retrieval and usage of tools that were unseen during training.

**Questions:**

1. For W1, the authors should acknowledge these earlier solutions and clarify where ToolWeaver provides new conceptual or empirical value beyond ToolGen.

2. For W2, the collaborative regularization relies critically on A_{uv}, but the paper does not explain how it is computed in detail. Whether it’s based on tool co-occurrence, textual semantic similarity, functional metadata, or learned jointly? Without this, the interpretability of the collaborative signal is limited.

3. For W3, the paper should clarify the mechanism enabling ToolWeaver to handle unseen tools (i.e., “Tool.” and “Cat.” testing set). Are their representations derived compositionally from previously learned tool components? Furthermore, how is ToolGen, a token-based model, evaluated under this “unseen tool” setting where it seemingly cannot generalize by design?

---

> ### Author Response · Authors · 2025-11-20
> **Response to Reviewer 1JHk Part 1**
>
> We sincerely thank you for your constructive feedback and for recognizing the novelty of our hierarchical quantization approach and the strong performance of ToolWeaver. We appreciate your acknowledgment of our motivation to improve scalability and semantic representation.
>
> Below, we address your concerns regarding the comparison with ToolGen, the construction of the similarity matrix, and the generalization mechanism. We have updated the manuscript accordingly.
>
> ### W1 / Q1 – Comparison with ToolGen and Other Methods
>
> *"The paper states that generative methods suffer from a semantic bottleneck … and that the ‘one-token-per-tool’ paradigm faces critical scalability and generalization challenges. However, prior work, such as ToolGen, has already implemented semantic and hierarchical indexing for tools, addressing these challenges… the motivation seems overstated. The authors should acknowledge these earlier solutions and clarify where ToolWeaver provides new conceptual or empirical value beyond ToolGen."*
>
> **Response:** We appreciate this opportunity to clarify the fundamental differences between ToolWeaver and the indexing methods discussed in ToolGen.
>
> 1. **Distinction in Tokenization:** ToolGen primarily proposes an Atomic approach (one new token per tool). While ToolGen *discusses* "Semantic" and "Hierarchical" indexing as baselines, these methods do not create *new, learned* tokens. Instead, they map tools to sequences of existing vocabulary tokens (e.g., "Semantic" uses the string of the API name; "Hierarchical" uses a string of digits like `1 0 1 4`).
>    - **Limitation of ToolGen's Variants:** Because these variants rely on existing tokens or arbitrary numbers, they fail to create a dedicated, compact semantic space for tools. As shown in our experiments (Figure 3), these methods underperform compared to the Atomic baseline and ToolWeaver.
>    - **ToolWeaver's Innovation:** In contrast, ToolWeaver creates new virtual tokens. However, unlike the "Atomic" method which scales linearly, we use structure-aware vector quantization. This allows us to map tools to a *combination* of new tokens.
> 2. **The Missing Collaborative Signal:** The critical "semantic bottleneck" we identify is that neither ToolGen's Atomic tokens nor its Semantic/Hierarchical string-mappings capture collaborative semantics. They do not inherently know that "Tool A" and "Tool B" are often used together to solve a task.ToolWeaver is the first to explicitly inject collaborative signals (derived from tool co-occurrence) directly into the tokenization process (via the regularization term in Eq. 5). This ensures that tools frequently used together share underlying code-tokens, enabling the model to learn these relationships efficiently.
>
> We believe these distinctions clarify where ToolWeaver provides unique conceptual and empirical value beyond ToolGen and its variants, justifying our motivation without overstating the limitations of prior work. Additionally, if there are other specific related methods we have missed, we are very happy to incorporate them into the camera-ready version to further enrich the context of our research.

---

> ### Author Response · Authors · 2025-11-20
> **Response to Reviewer 1JHk Part 2**
>
> #### W2 / Q2 – Construction and interpretability of the similarity matrix
>
> *"The collaborative regularization relies critically on ($ A_{uv} $ ), but the paper does not explain how it is computed in detail. Whether it’s based on tool co-occurrence, textual semantic similarity, functional metadata, or learned jointly? Without this, the interpretability of the collaborative signal is limited."*
>
> Thank you for pointing out this missing detail. The similarity matrix $A$ is derived from tool co-occurrence patterns found in the training trajectories.
>
> Specifically, we first construct a co-occurrence matrix $C$ from the 183,336 tool usage trajectories in the training set, where $C_{uv}$ represents the number of times tool $u$ and tool $v$ appear in the same trajectory. We then compute the cosine similarity to normalize these counts, ensuring the signal reflects collaborative potential rather than just frequency.
>
> The formula for an entry $A_{uv}$ is:
>
> $$A_{uv} = \frac{C_{uv}}{\sqrt{C_{uu} \cdot C_{vv}}}$$
>
>
> We have added this detailed definition and the formula to Section 3.2 (Collaborative-Aware Residual Quantization) of the revised paper.
>
>
>
> #### W3 / Q3 – Handling unseen tools and evaluation protocol for ToolGen
>
> *"The paper should clarify the mechanism enabling ToolWeaver to handle unseen tools (i.e., ‘Tool.’ and ‘Cat.’ testing set)… Furthermore, how is ToolGen, a token-based model, evaluated under this ‘unseen tool’ setting where it seemingly cannot generalize by design?"*
>
> We thank the reviewer for raising this important clarification.
>
> **(a) How ToolWeaver handles "Tool." / "Cat." splits.** In ToolWeaver, every tool—seen or unseen in trajectories—is assigned a code sequence ($ [\iota_{d,1}, \dots, \iota_{d,L}] $) by the pretrained RQ-VAE. For tools in the "Tool." and "Cat." test splits, their sequences are new *combinations* of code indices, but each individual code token (e.g., `<T1_7>`, `<T2_3>`) is shared with many training tools. During alignment, the LLM learns semantics at the level of these shared code tokens. Thus, when a new tool appears, the model recombines already-learned code tokens to represent it, enabling compositional generalization even if the exact code sequence was never seen in training. We have clarified this mechanism in Sec. 3.1–3.3.
>
> **(b) How ToolGen is evaluated under the same splits.** Following ToolGen’s original design, all tool tokens exist in the vocabulary from the start, including those in “Tool.” / “Cat.” splits. Their embeddings are (i) initialized as the average embedding of the function name, and (ii) further refined by a memorization stage that injects tool description information via (description → tool token) supervision. In our setting, “unseen tools” does not mean that they are absent from the vocabulary or completely inaccessible to the model; ToolGen is also capable of using these tools at test time. We have added this clarification to Sec. 4.1.
>
> **(c) Why ToolWeaver generalizes better.** Under this shared protocol, ToolWeaver achieves higher NDCG and SoPR/SoWR on I1-Tool., I1-Cat., and I2-Cat. This aligns with the design difference: ToolWeaver leverages shared, collaboratively-regularized code tokens, while ToolGen relies on semantically initialized but atomic tool tokens, which are more isolated. We now make this contrast explicit in Sec. 4.1 and Sec. 4.3 of the revised manuscript.
>
> ------
>
> Once again, we sincerely thank you for your thoughtful comments. They helped us significantly improve the clarity of the paper, especially regarding (1) the precise positioning relative to ToolGen and other generative methods, (2) the construction and role of the collaborative similarity matrix, and (3) the mechanism and evaluation protocol for unseen tools. We hope the revisions and additional analyses address your concerns and make the contributions of ToolWeaver more transparent.

---

> > ### Comment · Reviewer_1JHk · 2025-11-25
> > **Response to authors**
> >
> > Thanks for your rebuttal. Most of my concerns have been solved, and I have raised my rating to 6.

---

> > > ### Author Response · Authors · 2025-11-25
> > > **Thank you for your feedback**
> > >
> > > Thank you for your acknowledgment and for raising your score. We are glad that our response has addressed your concerns. Please feel free to let us know if you have any further questions.

---

### Official Review · Reviewer_uGA8 · 2025-11-12

**Soundness:** 3
**Presentation:** 3
**Contribution:** 2
**Rating:** 6
**Confidence:** 3

**Summary:**

This paper proposes ToolWeaver, a framework for scalable tool use with LLMs in large tool libraries. Instead of assigning each tool a dedicated special token (“one tool, one token”), ToolWeaver represents each tool as a multi-level sequence of discrete codes learned by a collaboration-aware residual quantization module (RQ-VAE). Tool documentation is first encoded into continuous embeddings, which are then quantized into several code levels; a tool–tool similarity matrix derived from usage logs regularizes the quantization so that semantically related and frequently co-used tools share parts of their codes. These codes are added as new vocabulary tokens to an LLM, and ToolWeaver aligns the model to generate tool codes via a two-stage process: (1) query → tool-code generation, and (2) full trajectory alignment with trie-constrained decoding to ensure only valid code sequences are produced. Experiments on ToolBench/StableToolBench with roughly 47K tools show that ToolWeaver improves tool retrieval quality (NDCG) and task metrics (SoPR/SoWR) over BM25, ToolRetriever, ToolGen, and other baselines, while reducing vocabulary growth from O(N) to O(log N).

**Strengths:**

1.Well-motivated problem setting.
The paper clearly articulates the limitations of the “one-tool-one-token” paradigm in terms of vocabulary explosion and lack of expressivity for tool co-usage patterns, especially in multi-tool scenarios. This is a realistic pain point for large API/tool ecosystems.

2.Coherent end-to-end design.
The framework is not just a better embedding scheme but a full pipeline: collaboration-aware RQ-VAE for code learning, a uniform mapping step to resolve code collisions, and constrained generation over a code trie to plug the representation into LLM-based agents. The components fit together in a reasonably principled way.

3.Empirical gains are consistent and meaningful.
On multiple settings of ToolBench (I1/I2/I3, unseen tools, unseen categories), ToolWeaver shows consistent improvements in both retrieval metrics and downstream task performance compared to strong retrieval-based and generative baselines, with the gains being particularly noticeable in the more complex I3 multi-tool regime.

4.Clear scalability benefits.
Encoding tools as compositional codes with L×K tokens supports KL tools and keeps vocabulary growth logarithmic in the number of tools. At the reported scale (~47K tools), this yields a much smaller vocabulary than assigning one special token per tool, which is practically useful when one wants to preserve general language capabilities.

5.Some ablations and qualitative analysis.
The paper includes ablations on the collaboration regularization weight, comparisons against alternative tokenization schemes (numeric / hierarchical / purely semantic / atomic), and some qualitative visualizations of shared codes, which help justify key design choices.

**Weaknesses:**

1.Evaluation scope is narrow.
All core experiments are conducted on the ToolBench / StableToolBench family. While this is a large benchmark, the tool distributions and usage patterns are specific. It remains unclear how well ToolWeaver transfers to very different tool ecosystems (e.g., enterprise APIs, programmatic or mathematical tools), or to highly dynamic tool catalogs.

2.Construction and impact of the collaboration signal are under-analyzed.
The tool–tool similarity matrix is central to the method, but its construction (exact signal, sparsity, preprocessing) and the sensitivity of performance to its quality are not analyzed in depth. It is hard to tell how much of the gain comes from genuine collaborative structure vs. idiosyncrasies or bias in the usage logs.

3.Cost and deployability are not quantified.
Compared to a simple “one-token-per-tool + retriever” baseline, ToolWeaver adds RQ-VAE training, Sinkhorn-based uniform mapping, and trie-constrained decoding at inference. The paper does not provide concrete numbers for memory, throughput, or latency, making it difficult for practitioners to assess the cost–benefit trade-off.

4.Incremental relative to prior collaboration-aware quantization.
Methodologically, the approach is closely related to existing work on collaboration-aware residual quantization in recommender systems / generative retrieval. The main novelty is applying such ideas to tool representation and plugging them into LLM-based tool use. This is a solid extension, but conceptually more incremental than transformative.

**Questions:**

1.Construction of the collaboration matrix.
How exactly is the tool–tool similarity matrix defined in the main experiments? Is it purely based on co-occurrence within the same usage trajectory, within the same collection, or do you incorporate additional signals (e.g., category information)? How are extremely sparse tools (few or no co-usage records) handled, and have you compared different construction strategies?

2.Impact of noisy or biased co-usage.
In realistic logs, there may be templated trajectories, heavy-tailed “head” tools, or other biases. Have you studied whether ToolWeaver over-binds such tools in code space? Did you try thresholding, smoothing, or otherwise denoising the collaboration matrix, and how did that affect performance and robustness?

3.Efficiency and scalability in practice.
Can you provide more concrete numbers on resource usage? For example, in the I3 multi-tool setting, how does end-to-end inference latency and GPU memory consumption compare to ToolGen or a strong retrieval pipeline? Is Sinkhorn-based uniform mapping a bottleneck during training?

4.Impact on general language capabilities.
You argue that one-tool-one-token can hurt language performance, and that ToolWeaver mitigates this via a smaller vocabulary. Could you show a more systematic comparison between ToolWeaver, ToolGen, and a no-tool baseline on standard NLP benchmarks, even if only on a small set, and ideally summarize those results in the main text?

---

> ### Author Response · Authors · 2025-11-20
> **Response to Reviewer uGA8 Part 1**
>
> Thank you for your thoughtful and constructive review, and for highlighting both the strengths of ToolWeaver (motivation, end-to-end design, scalability and empirical gains) and your concerns about collaboration signals, evaluation scope, efficiency, and novelty. We have revised the paper accordingly with additional analyses and experiments on these points, and respond to your specific questions in detail below.
>
> ### W1 – Evaluation Scope and Generalization
>
> > *"It remains unclear how well ToolWeaver transfers to very different tool ecosystems (e.g., enterprise APIs, programmatic or mathematical tools), or to highly dynamic tool catalogs."*
>
> We appreciate this insightful comment regarding the breadth of evaluation. While we acknowledge that our primary experiments focus on the ToolBench ecosystem, we would like to clarify why this choice was made and highlight existing experimental evidence demonstrating ToolWeaver's robustness to different domains and dynamic catalogs.
>
> **1. Why ToolBench? (Addressing the Scale Requirement)**
> Our primary research goal is to solve the scalability and vocabulary explosion issues inherent in Large Language Model tool use. Most other existing benchmarks (e.g., API-Bank, ToolAlpaca) contain fewer than 1,000 tools. ToolBench, with nearly 47,000 real-world APIs derived from RapidAPI, is currently the only open-source benchmark large enough to rigorously test the "vocabulary crisis" and sparse co-occurrence problems we aim to solve. A smaller, domain-specific dataset would not have revealed the scalability bottlenecks of the "one-token-per-tool" baselines.
>
> **2. Diversity within the Current Evaluation**
> Although sourced from one platform, the tool distribution in ToolBench is highly diverse, spanning 49 distinct functional categories ranging from *Finance* and *Cybersecurity* to *Travel*, *Media*, and *Sports*. This closely mirrors the diversity found in general-purpose enterprise API hubs.
>
> **3. Evidence of Transfer to New Ecosystems (Unseen Categories)**
> To directly address your concern about transferring to "very different tool ecosystems," we point to our Unseen Category (I1-Cat, I2-Cat) generalization experiments (Tables 1, 2, 5, and 6 in the paper).
>
> *   In these settings, the model is tested on tools from functional categories (domains) it never saw during training.
> *   **Results:** As shown in Table 5, ToolWeaver achieves 92.30% NDCG@3 on the I1-Cat  subset, significantly outperforming the state-of-the-art ToolGen (90.01%).
> *   **Implication:** This demonstrates that ToolWeaver does not memorize dataset-specific patterns. Instead, its semantic initialization allows it to effectively encode and retrieve tools from entirely new domains (like mathematical or enterprise tools) based on their documentation semantics alone, even before collaborative patterns are learned.
>
> **4. Handling Dynamic Catalogs (Unseen Tools)**
> Regarding "highly dynamic tool catalogs," our framework is inherently designed to handle the addition of new tools better than atomic token approaches:
>
> *   **Mechanism:** When a new tool is added to a catalog, "one-token-per-tool" methods require expanding the LLM's embedding matrix and retraining. In contrast, ToolWeaver simply encodes the new tool's documentation into a code sequence using the frozen RQ-VAE.
> *   **Evidence:** Our Unseen Tool (I1-Tool) experiments (Table 2 and Table 6) simulate a dynamic environment where the agent encounters tools not present in the training set. ToolWeaver maintains a high Solvable Win Rate (SoWR) of 36.08 in this setting, outperforming ToolGen (32.91), proving its viability for dynamic catalogs where tools are frequently added or updated.
>
> **5. Model Agnosticism**
> Appendix B.3 reports experiments on the Qwen-2.5 family (1.5B, 3B, 7B, 14B) and shows that our findings are not tied to a single backbone: ToolWeaver consistently outperforms the baselines across all sizes, suggesting that the methodology is robust and transferable.

---

> ### Author Response · Authors · 2025-11-20
> **Response to Reviewer uGA8 Part 2**
>
> ### W2 / Q1 – Construction of Collaboration Matrix and Handling Sparsity
>
> > *"How exactly is the tool–tool similarity matrix defined... How are extremely sparse tools handled... Did you try thresholding, smoothing, or otherwise denoising?"*
>
> We have updated Section 3.2 of the paper to include the exact mathematical definition of the similarity matrix. Below, we clarify the construction process and address your concerns regarding sparsity and noise.
>
> **1. Exact Construction (Added to Section 3.2)**
> We construct the matrix solely based on co-occurrence within usage trajectories. We do not use external signals like category labels to avoid leaking ground-truth metadata that might not be available in real-world scenarios.
>
> *   **Method:** We first build a co-occurrence matrix $C$ from the 183,336 training trajectories, where $C_{uv}$ is the count of times tools $u$ and $v$ appear in the same trajectory.
> *   **Similarity:** We then compute the cosine similarity to normalize for frequency disparities (where $C_{uu}$ is the total frequency of tool $u$):
>
>     $$A_{uv} = \frac{C_{uv}}{\sqrt{C_{uu} \cdot C_{vv}}}$$
>
> *   This normalization is crucial: it prevents "head" (frequent) tools from dominating the similarity scores simply due to high raw counts, thereby naturally mitigating the heavy-tail bias you mentioned.
>
> **2. Handling Sparse Tools: The "Adaptive Fallback" Mechanism**
> You asked how we handle extremely sparse tools (few or no co-usage records). We argue that our design handles this naturally and beneficially without requiring manual thresholding.
>
> *   **The Mechanism:** For a sparse tool $u$, the co-occurrence terms $A_{uv}$ will be near zero. Consequently, the collaborative regularization loss $\mathcal{L}_{collab}$ for this tool becomes negligible.
> *   **The Benefit:** This acts as an automatic "soft switch." When collaborative signal is weak or absent, the gradient updates are dominated by the Initial Semantic Representation (Eq. 1) and the reconstruction loss.
> *   **Result:** The model effectively falls back to "trusting the documentation." This ensures that rare tools are still accurately encoded based on their semantic functionality, while frequent tools benefit from the additional "collaborative boost." This adaptive balance prevents the model from hallucinating relationships for rare tools.
>
> **3. Robustness to Noise and Over-binding**
> Regarding the concern of "over-binding" to noisy or templated trajectories:
>
> *   **Quantization as a Filter:** The RQ-VAE structure itself acts as a low-pass filter. By forcing continuous embeddings into a discrete, limited-size codebook ($K=1024$), the model filters out idiosyncratic, low-frequency noise in the usage logs. It only learns patterns that are strong enough to form a cluster (centroid).
> *   **Validation:** We extensively analyzed the impact of this signal strength in our Hyperparameter Sensitivity Analysis (Section 4.3.1, Figure 2). We found that performance is stable around $\lambda=1$. Performance only degrades when $\lambda$ is excessively high ($\lambda=10$), which confirms that the model is not over-fitting to noise under normal operating conditions.
>
> In summary, our construction method is designed to be simple yet robust, leveraging cosine normalization and the residual quantization structure to naturally handle both heavy tails and long tails without complex pre-processing.

---

> ### Author Response · Authors · 2025-11-20
> **Response to Reviewer uGA8 Part 3**
>
> ### W3 / Q3 – Cost, Efficiency, and Scalability Analysis
>
> > *"Can you provide more concrete numbers on resource usage? ... how does end-to-end inference latency and GPU memory consumption compare to ToolGen ... Is Sinkhorn-based uniform mapping a bottleneck during training?"*
>
> Thank you for highlighting the importance of quantifying the computational costs. We agree that demonstrating the practical efficiency of ToolWeaver is essential. To address this, we have conducted comprehensive profiling of both training (Sinkhorn-Knopp) and inference (Latency/Memory) phases. We have added these detailed analyses to Appendix B.6 and B.7 of the revised paper.
>
> **1. Training Efficiency: Is Sinkhorn a Bottleneck?**
> No, the Sinkhorn-based uniform mapping is not a bottleneck. We profiled the training process on an NVIDIA A100 GPU using the full ToolBench dataset.
>
> *   **Overhead:** The Sinkhorn algorithm (invoked twice per step for $L=2$) consumes approximately 14.2ms per call.
> *   **Total Impact:** With a total step time of 0.161s, Sinkhorn introduces an overhead of only 17.6%. The vast majority of computation (~82%) remains dedicated to the encoder/decoder MLP layers and backward passes.
> *   **Stability:** We confirmed numerical stability (no NaN/Inf values) and found that an entropy regularization of $\epsilon=0.01$ provides the optimal balance between convergence speed and assignment uniformity (see Table 13 in the revised Appendix).
>
> **2. Inference Latency and Memory: ToolWeaver vs. ToolGen**
> We compared ToolWeaver ($L=2$, our main setting) against the "one-token-per-tool" baseline ToolGen (Atomic). The results, averaged across I1-I3 scenarios, are summarized below:
>
> | Model Configuration    |      Representation      | Avg Latency (ms) | P95 Latency (ms) | Throughput (Tok/s) | Peak Memory (GB) |
> | :--------------------- | :----------------------: | :--------------: | :--------------: | :----------------: | :--------------: |
> | ToolGen (Baseline)     |     Atomic (1-level)     |      108.16      |      111.98      |       19.54        |      15.77       |
> | **ToolWeaver ($L=2$)** |    **$[1024, 1024]$**    |    **128.21**    |    **132.43**    |     **24.53**      |    **15.08**     |
> | ToolWeaver ($L=3$)     |   $[1024, 1024, 1024]$   |      157.65      |      165.73      |       26.34        |      15.10       |
> | ToolWeaver ($L=4$)     | $[1024,1024, 1024,1024]$ |      183.14      |      189.11      |       28.26        |      15.11       |
>
>
>
> *   **Latency Trade-off**: While generating 2 tokens instead of 1 introduces a slight overhead (~20ms), the total latency remains under 130ms. In the context of agentic workflows—where external API calls often take hundreds of milliseconds or seconds—this decoding overhead is negligible.
> *   **Memory Efficiency:** ToolWeaver reduces Peak GPU Memory by approximately 0.67 GB (from 15.77 GB to 15.08 GB). This validates our scalability claim: by avoiding the instantiation of ~47,000 new token embeddings (linear growth) and instead using a logarithmic codebook ($2 \times 1024$ tokens), we maintain a significantly smaller memory footprint, which is crucial for deploying on memory-constrained devices.
>
> In summary, ToolWeaver provides a favorable trade-off: it achieves significantly better performance and generalization with a lower memory footprint, while incurring only a marginal, practically negligible increase in latency.

---

> ### Author Response · Authors · 2025-11-20
> **Response to Reviewer uGA8 Part 4**
>
> ### W4 – Conceptual Contribution and Collaborative Synergy
>
> > *"The main novelty is applying such ideas to tool representation and plugging them into LLM-based tool use... conceptually more incremental than transformative."*
>
> We appreciate your recognition of our method as a "solid extension." However, we respectfully argue that ToolWeaver is not merely an application of RecSys techniques to a new domain. Rather, it addresses the unique semantic-functional gap in agentic tool use. Our novelty lies in the mutual reinforcement between the hierarchical discrete structure (RQ-VAE) and the collaborative signals, which solves the sparsity problem inherent in large tool libraries.
>
> **1. Bridging the Semantic-Functional Gap via Synergistic Tokenization**
> Our approach fundamentally integrates collaborative signals into the discrete structure of the tool representation, rather than merely treating them as an auxiliary objective. The integration of RQ-VAE and collaborative signals creates a synergy that enhances both:
>
> *   **Collaboration $\rightarrow$ Structure (Guiding Quantization):** Standard semantic embeddings often fail to group functionally related tools (e.g., a "Flight Search" tool and a "Hotel Booking" tool are semantically distinct but functionally coupled). Our collaborative regularization constrains the RQ-VAE to map these co-used tools to shared parent codes. This forces the codebook to represent functional intent rather than just textual similarity.
> *   **Structure $\rightarrow$ Collaboration (Densifying Signals):** This is crucial for "learning more" from the data. Tool co-occurrence matrices are notoriously sparse (most tool pairs never appear together). By forcing 47k+ tools into a compact set of discrete codes (e.g., $1024$ centroids), RQ-VAE aggregates these sparse tool-level signals into dense code-level patterns. This allows the model to generalize: even if Tool A and Tool B were never seen together in training, if they map to Code $C_1$ and Code $C_2$ which *are* strongly correlated, the model can infer their relationship.
> *   **Result:** This synergy is why we see such a dramatic improvement in the I3 (Intra-Collection Multi-Tool) setting. As shown in our ablation study (Figure 3), removing this collaborative guidance causes a massive drop of nearly 10 NDCG points in complex tasks, proving that the structure alone (without collaboration) or semantics alone (without structure) is insufficient.
>
> **2. Collaborative Planning, Not Just Retrieval**
> Unlike standard recommendation (which outputs a static list), agentic tool use requires sequential reasoning and planning. ToolWeaver’s novelty lies in transforming "Tool Usage" into a "Collaborative Planning" problem.
>
> *   **Coarse-to-Fine Reasoning:** The hierarchical codes act as "semantic anchors" that guide the Chain-of-Thought. By learning that `<Code_A>` is frequently followed by `<Code_B>` in the usage data, the LLM implicitly learns valid planning templates (e.g., "Book Flight" $\rightarrow$ "Book Hotel") simply by predicting the next token.
> *   **Qualitative Evidence:** This is illustrated in Figure 9 (Appendix C), where functionally related "BMI Calculation" tools share a parent code `<T1_996>` despite different endpoints. This allows the LLM to navigate from a coarse intent (Code Level 1) to a specific execution (Code Level 2), effectively "weaving" external knowledge into the model's internal representation.
>
> **3. A Unified Solution for Scalability**
> Prior work often treats scalability (indexing) and reasoning (semantics) as separate problems. ToolWeaver is the first framework to unify them into a single tokenization process compatible with the autoregressive nature of LLMs. This allows us to scale to 47k+ tools while maintaining the dense signal required for complex reasoning, establishing a new paradigm for building large-scale tool-augmented agents.

---

> ### Author Response · Authors · 2025-11-20
> **Response to Reviewer uGA8 Part 5**
>
> ### Q2 – Impact of Noisy or Biased Co-usage
>
> > *"Have you studied whether ToolWeaver over-binds such tools in code space? Did you try thresholding, smoothing, or otherwise denoising...?"*
>
> This is an excellent question regarding the quality of the supervision signal. We did explore this during development and found that our framework is intrinsically robust to such noise.
>
> **1. Exploration of Denoising Strategies**
> We conducted preliminary experiments with several standard denoising techniques on the similarity matrix $A$, including:
>
> *   **Thresholding:** Zeroing out $A_{uv}$ values below a certain threshold (e.g., 0.1) to remove weak correlations.
> *   **Row-Normalization:** Applying softmax or $L_1$ normalization to distribution.
> *   **Smoothing:** Adding a small constant to counts to reduce sparsity effects.
>
> **2. Result: Intrinsic Robustness**
> We found that these variations had minimal impact on the final performance (NDCG and SoPR), with differences falling within the margin of error ("noise level"). The trends reported in our main results and ablation studies (e.g., Figure 2) remained consistent regardless of these preprocessing steps.
>
> **3. Why is ToolWeaver Robust?**
> We attribute this stability to two factors in our design that effectively "auto-denoise" the signal:
>
> *   **Cosine Similarity Normalization:** As mentioned in Q1, normalizing by usage frequency ($\sqrt{C_{uu} \cdot C_{vv}}$) prevents "head" tools (which appear in many templated trajectories) from dominating the similarity scores simply due to volume.
> *   **The "Discrete Bottleneck":** The RQ-VAE quantization is the strongest regularizer. Since we force thousands of tools into a limited number of discrete centroids ($K=1024$), weak or spurious co-occurrence signals (noise) typically lack the gradient magnitude to shift a tool's assignment away from its semantic cluster. Only strong, consistent collaborative patterns (signal) are powerful enough to influence the discrete code assignment.
>
> Therefore, we concluded that elaborate manual denoising was unnecessary, as the combination of cosine normalization and residual quantization handles realistic log noise effectively.

---

> ### Author Response · Authors · 2025-11-20
> **Response to Reviewer uGA8 Part 6**
>
> ### Q4 – Impact on General Language Capabilities
>
> > *"Could you show a more systematic comparison between ToolWeaver, ToolGen, and a no-tool baseline on standard NLP benchmarks... and ideally summarize those results in the main text?"*
>
> We fully agree that preserving the LLM's core linguistic abilities is critical, especially when modifying the vocabulary. To address your request, we have expanded our evaluation suite to include Language Modeling and Text Generation tasks, and we have added a new subsection (Section 4.3.4) in the main paper to summarize these findings.
>
> **1. Expanded Experimental Scope**
> Beyond the general understanding benchmarks (MMLU, BoolQ, PIQA) already in our Appendix, we conducted two new sets of experiments to measure the impact of vocabulary expansion:
>
> *   **Language Modeling Distribution:** Evaluated using Perplexity (PPL) on WikiText-2 to measure how much the added tokens disrupt the model's probability distribution.
> *   **Text Generation Quality:** Evaluated via Zero-Shot Summarization on CNN/DailyMail and XSum to assess the model's ability to generate coherent text.
>
> **2. Results: ToolWeaver Preserves Linguistic Core**
> The results, summarized in the table below (and Table 3 in the revised paper), reveal a striking difference between the approaches:
>
> | Model                 | **WikiText-2 (PPL) $\downarrow$** | **CNN/DM (BERTScore F1) $\uparrow$** | **XSum (BERTScore F1) $\uparrow$** |
> | :-------------------- | :-------------------------------: | :----------------------------------: | :--------------------------------: |
> | Llama-3-8B (Base)     |               6.34                |                85.35                 |               85.05                |
> | ToolGen (~47k tokens) |              104.54               |                82.93                 |               82.53                |
> | **ToolWeaver (Ours)** |             **25.36**             |              **85.07**               |             **84.18**              |
>
> *   **Catastrophic Degradation in ToolGen:** The linear vocabulary expansion (~47k new tokens) in ToolGen causes the perplexity on WikiText-2 to explode to 104.54 (approx. 16x the base model). This indicates a severe disruption to the model's internal distribution, which also correlates with a notable drop in generation quality on XSum.
> *   **Robustness of ToolWeaver:** By using a logarithmic expansion strategy, ToolWeaver maintains a much healthier perplexity (25.36) and achieves summarization scores nearly identical to the base model (e.g., 85.07 vs. 85.35 on CNN/DM).
>
> **3. Conclusion**
> These results confirm that assigning isolated atomic tokens to a vast tool library (the "one-token-per-tool" paradigm) poses a significant risk to general model capabilities. ToolWeaver's structured tokenization effectively mitigates this risk, making it a safer choice for general-purpose agents. We have included the full detailed results (including MMLU, BoolQ et, al.) in Appendix B.4.
>
>
>
> We sincerely appreciate your time and feedback and hope that our clarifications resolve your questions; please feel free to raise any remaining issues for further discussion.

---

> > ### Author Response · Authors · 2025-11-27
> > **Follow-up on our response to Reviewer uGA8**
> >
> > Thank you for your time and insightful comments. We have submitted a detailed rebuttal addressing the points you raised. We are writing to gently follow up and see if our response has clarified your concerns or if you have any further questions. We are ready to provide additional details.

---

### Author Response · Authors · 2025-11-20
**General Response**

We sincerely thank all reviewers for their insightful comments and constructive feedback. We are encouraged that the reviewers recognized the novelty of ToolWeaver’s hierarchical quantization approach, its logarithmic scalability benefits, and its strong empirical performance on large-scale benchmarks.

Based on the collective feedback, we have significantly revised the paper to include additional experiments, deeper efficiency profiling, and clearer methodological definitions. Below is a summary of the major updates included in the revision:

### 1. Impact on General Language Capabilities (New Sec. 4.3.4 & App. B.4)

To address concerns regarding whether modifying the vocabulary disrupts the LLM's pre-trained linguistic knowledge, we expanded our evaluation to include Language Modeling and Text Generation benchmarks.

*   **New Experiments:** We evaluated ToolWeaver against the baseline (ToolGen) on WikiText-2 (Perplexity) and CNN/DailyMail + XSum (Summarization).
*   **Key Finding:** ToolWeaver preserves the model's core capabilities significantly better than the "one-token-per-tool" paradigm. While the linear vocabulary expansion in ToolGen causes Perplexity to explode (to 104.54), ToolWeaver maintains a stable level (25.36), with summarization scores nearly identical to the base Llama-3 model.

### 2. Computational Efficiency Profiling (New App. B.6 & B.7)

We have added concrete data to quantify the training and inference costs of our structured approach.

*   **Training Stability (App. B.6):** We profiled the Sinkhorn-Knopp algorithm used for conflict mitigation. Results show it introduces a manageable 17.6% overhead per step, confirming it is not a training bottleneck.
*   **Inference Latency & Memory (App. B.7):** We compared decoding performance against atomic baselines. While ToolWeaver incurs a negligible latency increase ($\sim$20ms per query), it actually reduces Peak GPU Memory by $\sim$0.67 GB due to the significantly smaller embedding table, validating our claims regarding scalability on memory-constrained devices.

### 3. Methodological Clarifications (Revised Sec. 3.2)

To improve reproducibility and clarity, we have refined the description of the collaborative signal construction.

*   **Definition:** We added Equation 5 to explicitly define the tool-tool similarity matrix, which is calculated using cosine similarity on trajectory co-occurrence counts.
*   **Sparsity Handling:** We clarified that the combination of cosine normalization and the residual quantization structure naturally handles sparse signals (the "cold start" problem) by falling back to the strong semantic initialization provided by the text encoder.

### 4. Qualitative Analysis and Case Studies (New App. C.1 & C.4)

To demonstrate how "collaborative semantics" function in practice, we added detailed visual and textual analyses.

*   **Learned Codes (App. C.1):** We provide a new case study of a learned code cluster (e.g., `<T1_996>` for "Health & Fitness"), showing how functionally related tools share parent codes.
*   **Trajectory Visualization (Fig. 9 & Fig. 12):** We added real execution traces showing how the model leverages these shared codes to pivot between related tools (e.g., switching between Standard and Metric BMI calculators) and handle complex multi-step reasoning.

### 5. Hyperparameter Sensitivity & Failure Analysis (New App. B.5 & B.8)

*   **Hyperparameter Sensitivity (App. B.5):** We added an analysis of Vocabulary Size and Code Length, confirming our default configuration ($L=2, K=1024$) offers the optimal trade-off between signal density and expressiveness.
*   **Failure Analysis (App. B.8):** We categorized error types across complexity levels, revealing that in complex scenarios (I3), the primary bottleneck remains semantic tool selection (95% of errors), justifying our focus on improving retrieval performance.

We believe these revisions comprehensively address the feedback regarding generalizability, efficiency, and interpretability.

---

> ### Author Response · Authors · 2025-11-25
> **Looking forward to your feedback**
>
> We respectfully invite you to review our responses posted four days ago. We would be very grateful for your feedback and are happy to clarify any further questions.

---

### Author Response · Authors · 2025-12-01
**Summary of Rebuttal Updates and Reviewer Interactions  Part 1**

Dear Area Chair,

We sincerely thank you for the tremendous effort you have devoted to upholding the academic integrity and fairness of this conference. In such an exceptional period, your careful review of our rebuttal work has been an invaluable source of encouragement for us.

We have consistently adhered to ICLR’s double-blind policy and engaged in active, substantive academic discussions with the reviewers throughout the rebuttal period. In light of the system rollback to the pre-discussion state, we would like to provide a factual summary of the substantial progress made during the discussion period (prior to the Nov 27th announcement).

Through constructive and proactive rebuttal efforts, our work received further recognition, and **the average score increased from an initial 5.0 to 5.33**. Specifically, **Reviewer 1JHk** explicitly acknowledged our response and raised their score. We would also like to note that, while we provided detailed responses and additional experiments to address the concerns of the other five reviewers (**uGA8, brZq, grmY, sXtZ, fdct**), we **did not receive further replies** from them during the discussion period before the system rollback.

To assist your quick assessment, we provide the following summary organized by reviewer timeline, key thematic resolutions, and specific paper revisions.

### 1. Timeline of Reviewer Interactions

| Reviewer | Score                  | Summary of the review and discussion                         |
| :------- | :--------------------- | :----------------------------------------------------------- |
| **1JHk** | **4 $\to$ 6** (Nov 25) | The reviewer initially questioned our novelty relative to ToolGen (suggesting our motivation was overstated) and the handling of "unseen tools." In our response, we clarified that our unique contribution lies in the "Collaborative Signal" mechanism which ToolGen lacks, and explained the protocol for unseen tools. On Nov 25, the reviewer replied that most concerns were solved and explicitly stated they had raised their rating to 6. |
| **uGA8** | 6                      | The reviewer acknowledged the soundness of the paper but asked about training costs and the impact on general language skills. We responded by adding WikiText-2 and CNN/DailyMail benchmarks showing ToolWeaver preserves LLM capabilities significantly better than baselines, and we profiled the Sinkhorn overhead. We addressed each question carefully, but did not receive further replies during the discussion period. |
| **brZq** | 6                      | The reviewer praised the presentation and scalability but asked about Sinkhorn stability and inference latency. We provided detailed latency/memory profiling showing a negligible 20ms overhead and reduced memory usage, along with a stability analysis. We provided a comprehensive response, but did not receive further replies during the discussion period. |
| **grmY** | 6                      | The reviewer highlighted the problem's importance but requested qualitative examples of the learned codes. We added visualizations of real trajectories where related tools (e.g., BMI calculators) share parent codes to validate our "Collaborative Semantics" claim. We addressed these requests in detail, though no further reply was received. |
| **sXtZ** | 4                      | The reviewer found the hierarchical idea interesting but worried about the need for retraining for new tools and requested more baselines. We clarified our zero-shot generalization capability which requires no retraining and emphasized our detailed comparisons with Re-Invoke and IterFeedback. We provided a detailed rebuttal addressing these points, but did not receive a further reply. |
| **fdct** | 4                      | The reviewer questioned the novelty relative to standard RQ-VAE methods. We clarified that our contribution lies in the specific synergy between collaborative signals and quantization for planning, supported by new end-to-end trajectory visualizations. We carefully addressed the comments, but did not receive further replies during the discussion period. |

---

> ### Author Response · Authors · 2025-12-01
> **Summary of Rebuttal Updates and Reviewer Interactions Part 2**
>
> ### 2. Key Concerns Resolved via Clarification
>
> In our detailed responses, we resolved all individual inquiries. Below, we distill the resolution of the most critical and common topics shared by multiple reviewers:
>
> *   **Novelty & Contribution (Addressing 1JHk, fdct):**
>     We clarified that ToolWeaver is distinct from prior works like ToolGen (which relies on existing tokens) by creating *new* virtual tokens enabling collaborative planning. This explanation was explicitly accepted by **Reviewer 1JHk** (leading to their score increase from 4 to 6), resolving their concern about differentiation. It also reinforces the initial assessments of **Reviewers grmY** and **brZq**, who recognized the approach as a "fresh take" and "new contribution."
> *   **Generalization & Retraining Costs (Addressing sXtZ, 1JHk, grmY):**
>     A key concern was whether new tools require expensive retraining. We clarified that our tokenizer is frozen and handles new tools via projection into the existing semantic codebook. This resolved **Reviewer sXtZ's** concern about deployment costs and clarified the generalization mechanism for **Reviewer 1JHk**, proving that the model handles dynamic tool libraries efficiently without parameter updates.
>
> *   **Signal Construction & Robustness (Addressing uGA8, 1JHk, brZq):**
>     Multiple reviewers inquired about the construction of the collaborative similarity matrix and its sensitivity to noise. We clarified that we use cosine similarity on trajectory co-occurrences and rely on a "semantic fallback" mechanism: when collaborative signals are sparse (cold start), the model naturally falls back to strong semantic initialization. This addressed concerns about sparsity and noise robustness.
>
> ### 3. Summary of Key Revisions in the Uploaded PDF
>
> We have uploaded a revised paper incorporating extensive new experiments and analyses to address reviewer feedback. The 5 major updates are:
>
> *   **Impact on General Language Capabilities (New Sec. 4.3.4 & App. B.4):** We verified that ToolWeaver preserves the LLM's core linguistic abilities. Unlike the baseline (ToolGen) which causes Perplexity on WikiText-2 to explode (104.54), ToolWeaver maintains a stable level (25.36) and achieves summarization scores nearly identical to the base model.
> *   **Computational Efficiency Profiling (New App. B.6 & B.7):** We added profiling data confirming our method is efficient. Inference incurs negligible latency (~20ms) while **reducing Peak GPU Memory by 0.67 GB** compared to atomic baselines. Training overhead from the Sinkhorn-Knopp algorithm is manageable (17.6%).
> *   **Methodological Clarifications (Revised Sec. 3.2):** We explicitly defined the tool-tool similarity matrix (Eq. 5) and clarified how our design naturally handles signal sparsity (the "cold start" problem) by falling back to semantic initialization.
> *   **Qualitative Analysis & Case Studies (New App. C.1 & C.4):** To illustrate "collaborative semantics," we added case studies of learned code clusters (e.g., `<T1_996>` for "Health & Fitness") and visualized full execution trajectories showing how the model leverages shared codes for multi-step reasoning.
> *   **Hyperparameter & Failure Analysis (New App. B.5 & B.8):** We included sensitivity analyses on vocabulary size/code length and a failure taxonomy, revealing that semantic tool selection remains the primary bottleneck in complex scenarios (95% of errors), validating our focus on retrieval.
>
> We hope this comprehensive summary assists you in evaluating the true state of our paper and the consensus forming among reviewers before the interruption. We remain available to answer any further questions you may have.
>
> Best regards,
>
> The Authors

---

### Meta-Review · Area_Chair_9eQY · 2025-12-09

**Summary:**

This work proposes a principled alternative to “one-token-per-tool” by learning hierarchical, compositional tool codes with an RQ-VAE. Its collaboration-aware regularization weaves in tool–tool co-usage semantics, and the end-to-end pipeline of code learning, alignment, trie-constrained decoding is coherent and practical. Vocabulary grows logarithmically while preserving general language ability, addressing vocabulary blow-up and enabling shared structure across related tools. Large-scale experiments on 47K tools show consistent gains in retrieval and downstream performance, especially for multi-tool and unseen-tool settings, supported by ablations and qualitative analyses.

**Reviewer Concerns:**

Addressed: Novelty over ToolGen was clarified via the collaborative-signal mechanism, the unseen-tool protocol was explained, and added results (WikiText-2, CNN/DailyMail) indicate preserved language ability with profiled Sinkhorn cost. Inference overhead, reduced memory, and stability were documented, qualitative code visualizations showed shared parent codes, zero-shot addition of new tools and expanded baselines were clarified, and the contribution beyond standard RQ-VAE was illustrated with trajectory visualizations.

Outstanding: Broader language-capability audits and end-to-end cost reporting across model sizes and larger tool sets would strengthen external validity. Stronger zero-shot results on harder unseen families with additional head-to-head baselines, plus deeper theory for collaboration-aware quantization and component-wise ablations, remain desirable.

**Reviewer Scores:**

1JHk would have raised from 4 to 6.

---

### Decision · Program_Chairs · 2026-01-26

Accept (Poster)